# AHA: Human-Assisted Out-of-Distribution Generalization and Detection

**Haoyue Bai, Jifan Zhang, Robert Nowak**
University of Wisconsin-Madison
{baihaoyue, jifan}@cs.wisc.edu, rdnowak@wisc.edu

## Abstract

Modern machine learning models deployed often encounter distribution shifts in real-world applications, manifesting as covariate or semantic out-of-distribution (OOD) shifts. These shifts give rise to challenges in OOD generalization and OOD detection. This paper introduces a novel, integrated approach AHA (**A**daptive **H**uman-**A**ssisted OOD learning) to simultaneously address both OOD generalization and detection through a *human-assisted framework* by labeling data in the wild. Our approach strategically labels examples within a novel maximum disambiguation region, where the number of semantic and covariate OOD data roughly equalizes. By labeling within this region, we can maximally disambiguate the two types of OOD data, thereby maximizing the utility of the fixed labeling budget. Our algorithm first utilizes a noisy binary search algorithm that identifies the maximal disambiguation region with high probability. The algorithm then continues with annotating inside the identified labeling region, reaping the full benefit of human feedback. Extensive experiments validate the efficacy of our framework. We observed that with only a few hundred human annotations, our method significantly outperforms existing state-of-the-art methods that do not involve human assistance, in both OOD generalization and OOD detection. Code is publicly available at `https://github.com/HaoyueBaiZJU/aha`.

## 1 Introduction

Modern machine learning models deployed in the real world often encounter various types of distribution shifts. For example, out-of-distribution (OOD) covariate shifts arise when the domain and environment of the test data differ from the training data. OOD semantic shifts occur when the model encounters novel classes during testing. This gives rise to two important challenges: OOD generalization [2, 1, 102], which addresses distribution mismatches between training and test data related to covariate shifts, and OOD detection [46, 62, 90], which aims to identify examples from semantically unknown categories that should not be predicted by the classifier, relating to semantic shifts. The natural coexistence of these different distribution shifts in real-world scenarios motivates the simultaneous handling of both tasks, a direction that has not been largely explored previously, as most existing approaches are highly specialized in one task.

Specifically, we consider a generalized characterization of the wild data setting [6] that naturally arises in the model's operational environment:

$$\mathbb{P}_{\text{wild}} := (1 - \pi_s - \pi_c)\mathbb{P}_{\text{in}} + \pi_c \mathbb{P}_{\text{out}}^{\text{covariate}} + \pi_s \mathbb{P}_{\text{out}}^{\text{semantic}},$$

where $\mathbb{P}_{\text{in}}$ denotes the marginal distributions of in-distribution (ID) data, $\mathbb{P}_{\text{out}}^{\text{covariate}}$ represents covariate-shifted OOD data, and $\mathbb{P}_{\text{out}}^{\text{semantic}}$ indicates semantic-shifted OOD data. This is challenging as we lack access to both the category labels and distribution types of this wild mixture data, which is crucial for OOD learning. To tackle this challenge, it is natural to develop a human-assisted framework and

38th Conference on Neural Information Processing Systems (NeurIPS 2024).

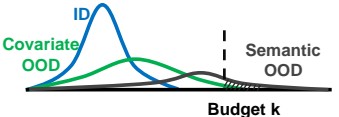

(a) Top-$k$ most OOD examples

(b) Near-boundary region

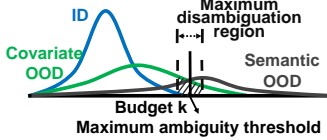

(c) Maximum disambiguation region (Ours)

Figure 1: Illustration and comparison of three different labeling regions. The horizontal axis is the OOD score, and the vertical axis is the frequency. Note that we color the three different sub-distributions (ID, covariate OOD, semantic OOD) separately for clarity. In practice, the membership is not revealed on these unlabeled wild data.

selectively label a set of examples from the wild data distribution. These examples are then used to train a multi-class classifier and an OOD detector. A critical yet unresolved question thus arises:

*By leveraging human feedback, can we identify and label a small set of examples that significantly enhances both OOD generalization and detection?*

In this paper, we propose the first algorithm AHA (**A**daptive **H**uman-**A**ssisted OOD learning) that incorporates human assistance in improving both OOD generalization and detection together. Given a limited labeling budget, our strategy selects wild examples that predominantly exhibit covariate shifts or semantic shifts, as these are the most informative for improving a model's OOD generalization and detection performances. At the core of our approach, we identify a novel labeling region, *the maximum disambiguation region*. Within this region, the densities of covariate OOD and semantic OOD examples approximately equalize, making it difficult for the OOD detector to differentiate between the two types of OOD data.

As demonstrated in Figure 1(c), the maximum disambiguation region is centered around the *maximum ambiguity threshold*, where the densities of the two types of OOD data exactly equalize. By labeling examples around this threshold, we therefore also maximize the total human corrections to the given OOD detector. Naturally, our algorithm invokes a two-phased procedure. First, we address the challenge of identifying the maximum ambiguity threshold by framing it as a noisy binary search problem and utilize an off-the-shelf adaptive labeling algorithm [56]. For the second phase, we label equal number of examples adjacent to the identified threshold from both sides.

Extensive experiments demonstrate the efficacy of AHA for both OOD generalization and detection. We observe that with only a few hundred human annotations, AHA notably improves OOD generalization and detection over existing SOTA methods that do not involve human assistance. Compared to most related literature [6], our findings indicate that obtaining just a few hundred human labels can reduce the average OOD detection error by 15.79% in terms of FPR95, and increase the accuracy of neural networks on covariate OOD data by 5.05% (see Table 1).

Table 1: **Results highlight**: comparison with state-of-the-art method SCONE on CIFAR-10 benchmrk.

| Method | SVHN $\mathbb{P}_{out}^{semantic}$, CIFAR-10-C $\mathbb{P}_{out}^{covariate}$ | | | | LSUN-C $\mathbb{P}_{out}^{semantic}$, CIFAR-10-C $\mathbb{P}_{out}^{covariate}$ | | | | Texture $\mathbb{P}_{out}^{semantic}$, CIFAR-10-C $\mathbb{P}_{out}^{covariate}$ | | | |
| | OOD Acc.↑ | ID Acc.↑ | FPR↓ | AUROC↑ | OOD Acc.↑ | ID Acc.↑ | FPR↓ | AUROC↑ | OOD Acc.↑ | ID Acc.↑ | FPR↓ | AUROC↑ |
|---|---|---|---|---|---|---|---|---|---|---|---|---|
| SCONE | 84.69 | 94.65 | 10.86 | 97.84 | 84.58 | 93.73 | 10.23 | 98.02 | 85.56 | 93.97 | 37.15 | 90.91 |
| **Ours** | **89.01** | 94.67 | **0.08** | **99.99** | **90.69** | 94.45 | **0.02** | **99.98** | **90.51** | 94.54 | **5.63** | **97.95** |

| Method | Places $\mathbb{P}_{out}^{semantic}$, CIFAR-10-C $\mathbb{P}_{out}^{covariate}$ | | | | LSUN-R $\mathbb{P}_{out}^{semantic}$, CIFAR-10-C $\mathbb{P}_{out}^{covariate}$ | | | | Average | | | |
| | OOD Acc.↑ | ID Acc.↑ | FPR↓ | AUROC↑ | OOD Acc.↑ | ID Acc.↑ | FPR↓ | AUROC↑ | OOD Acc.↑ | ID Acc.↑ | FPR↓ | AUROC↑ |
|---|---|---|---|---|---|---|---|---|---|---|---|---|
| SCONE | 85.21 | 94.59 | 37.56 | 90.90 | 80.31 | 94.97 | 0.87 | 99.79 | 84.95 | 94.32 | 19.33 | 95.49 |
| **Ours** | **88.93** | 94.30 | **11.88** | **95.60** | **90.86** | 94.32 | **0.07** | **99.98** | **90.00** | 94.46 | **3.54** | **98.70** |

Our key contributions are:

- We are the first to leverage human assistance in improving both OOD generalization and detection, offering a natural and effective approach for labeling wild data with heterogeneous data shifts.
- We propose a novel labeling strategy that targets the maximum disambiguation region, which significantly enhances both OOD generalization and detection when labeled.
- Extensive experiments and ablation studies demonstrate the effectiveness of our human-assisted method. AHA shows robust performance in both OOD generalization and detection.

## 2 Related Works

**Out-of-distribution generalization** is an important and challenging problem in machine learning, arising when there are distribution shifts between the training and test data. Compared to traditional

domain adaptation tasks [12, 36, 89, 57, 32, 99, 74], OOD generalization is more critical as it focuses on generalizing to covariate-shifted data distributions that are unseen during training [8, 61, 114, 71, 40, 69, 9, 58]. A primary set of approaches to OOD generalization involves extracting domain-invariant representations. Strategies include invariant risk minimization [2, 79, 111, 1], domain adversarial learning [65, 115, 86, 37, 66], meta-learning [63, 76], and others [78, 16, 7]. Other sets of approaches for OOD generalization include single domain generalization [75, 92], test-time adaptation [52, 110], and model ensembles [3, 77]. SCONE [6] aims to enhance OOD generalization and detection by leveraging unlabeled data from the wild. Based on the problem setting of SCONE, we propose to integrate human assistance to enhance OOD robustness and improve OOD detection accuracy. The primary motivation is to identify the optimal labeling regions within the wild data. We find that even a few hundred human-labeled instances, chosen based on our selection criteria, can significantly enhance performance for both tasks.

**Out-of-distribution detection** has gained increasing attention in recent years. There are primarily two sets of approaches to OOD detection: post hoc methods and regularization-based methods. Post hoc methods involve designing OOD scores at test time, which include confidence-based methods [46], energy-based scores [68, 109], gradient-based scores [10, 26], and distance-based scores [62, 91]. Another set of approaches involves leveraging training-time regularization for OOD detection by relying on an additional clean set of semantic OOD data [47, 44, 100]. Some recent studies propose utilizing wild mixture data for OOD detection. For example, WOODS [55] considers a wild mixture of both unlabeled ID and semantic OOD data. SCONE [6] includes a wild mixture of unlabeled ID, semantic OOD, and covariate OOD data, which are suitable for real-world scenarios. In contrast to previous work, we propose a human-assisted approach for the wild mixture setting and observe that only a few hundred human annotations can significantly improve robustness and OOD detection. Unlike [96], which utilizes adaptive human review for OOD detection via a fixed false positive rate threshold, our approach is fundamentally different. We collect informative examples to finetune the model and OOD detector, simultaneously improving OOD detection and generalization.

**Noisy binary search.** Our algorithm utilizes a noisy binary search algorithm to find the threshold where the density difference between covariate OOD and semantic OOD examples flip from negative to positive. In traditional binary search, one simply shrinks the possible interval of the threshold by half based on the observation of either a negative or a positive signal. However, in noisy binary search, the observations are inherently noisy with some probabilities. As a result, one necessarily needs to maintain a *high probability confidence interval* of where the threshold may be located. The noisy binary search problem has been primarily studied in combinatorial bandits [18, 34, 17, 15, 33, 54] and agnostic active learning [22, 41, 42, 23, 49, 53, 56]. We primarily utilize a version of the fix-budget algorithm from [56] as it is proven to be near instance-optimal. In the past, noisy binary search algorithms have been widely applied in applications such as text classification [84], wireless networks [87, 88] and training neural networks on in-distribution data [108, 73].

**Deep active learning** is a vital paradigm in machine learning that emphasizes the selection of the most informative data points for labeling, enabling efficient and effective model training with limited labeled data [107]. There are two main groups of algorithms: uncertainty sampling and diversity sampling. Uncertainty sampling aims to identify and select data examples where model confidence is low in order to reduce uncertainty when labeled [35, 28, 11, 97]. Diversity sampling aims to query a batch of diverse examples that are representative of the unlabeled pool for the overall data distribution [85, 39, 38, 112, 20]. Recently, some hybrid methods have arisen that consider both uncertainty sampling and diversity sampling, which query a batch of informative and diverse examples [5, 4, 20, 70]. Another line of work is deep active learning with class imbalance [21, 59, 31]. Some recent advances consider distribution shifts in the context of deep active learning [105, 13]. In this work, we consider OOD robustness and tackle the challenging scenario of unlabeled wild distributions, training a robust multi-class classifier and an OOD detector simultaneously.

## 3 Problem Setup

**Labeled in-distribution data.** Let $\mathcal{X}$ denote the input space and the label space $\mathcal{Y} := [K]$ consists of $K$ classes. We have access to an initial labeled training set of $M$ examples $\mathcal{S}_{\text{in}} \sim \mathbb{P}_{\mathcal{X}\mathcal{Y}}^M$.

**Unlabeled wild data.** When a model is deployed into a wild environment, it encounters unlabeled examples that exhibit various distributional shifts. We consider a generalized characterization of the

wild data as: $\mathbb{P}_{\text{wild}} := (1 - \pi_c - \pi_s)\mathbb{P}_{\text{in}} + \pi_c\mathbb{P}_{\text{out}}^{\text{covariate}} + \pi_s\mathbb{P}_{\text{out}}^{\text{semantic}}$, where $1 - \pi_c - \pi_s$, $\pi_c$ and $\pi_s$ are non-negative ratios, unknown to the learner.

- $\mathbb{P}_{\text{in}}$ refers to the ID data, which represents the marginal distribution of the initially labeled dataset.
- $\mathbb{P}_{\text{out}}^{\text{covariate}}$ represents the covariate OOD distribution (*OOD generalization*). The label space remains the same as in the training data, but the input space undergoes shifts in style and domain.
- $\mathbb{P}_{\text{out}}^{\text{semantic}}$ represents the semantic OOD distribution (*OOD detection*), which encompasses semantics outside the known categories $\mathcal{Y} := [K]$. These semantics should not be predicted by the model.

**Learning framework.** Let $f_{\mathbf{w}} : \mathcal{X} \mapsto \mathbb{R}^K$ denote a function for the classification task, which predicts the label of an input sample $\mathbf{x}$ as $\widehat{y}(f_{\mathbf{w}}(\mathbf{x})) := \arg\max_y f_{\mathbf{w}}^{(y)}(\mathbf{x})$. To detect the semantic OOD data, we train a ranking function $g_{\boldsymbol{\theta}} : \mathcal{X} \to \mathbb{R}$ with parameter $\boldsymbol{\theta}$. With the ranking function $g_{\boldsymbol{\theta}}$, one can define the OOD detector as a threshold function $D_{\boldsymbol{\theta}}(\mathbf{x}; \lambda) := \begin{cases} \text{ID} & \text{if } g_{\boldsymbol{\theta}}(\mathbf{x}) > \lambda \\ \text{OOD} & \text{if } g_{\boldsymbol{\theta}}(\mathbf{x}) \leq \lambda \end{cases}$. The threshold value $\lambda$ is typically chosen so that a high fraction of ID data is correctly classified.

**Learning goal.** We aim to evaluate our model based on the following measurements: (1) ID-Acc$(f_{\mathbf{w}})$ := $\mathbb{E}_{(\mathbf{x},y)\sim\mathbb{P}_{\mathcal{X}\mathcal{Y}}}(\mathbb{1}\{\widehat{y}(f_{\mathbf{w}}(\mathbf{x})) = y\})$; (2) OOD-Acc$(f_{\mathbf{w}})$ := $\mathbb{E}_{(\mathbf{x},y)\sim\mathbb{P}_{\text{out}}^{\text{covariate}}\cdot\mathbb{P}_{\mathcal{Y}|\mathcal{X}}}(\mathbb{1}\{\widehat{y}(f_{\mathbf{w}}(\mathbf{x})) = y\})$; (3) FPR$(g_{\boldsymbol{\theta}})$ := $\mathbb{E}_{\mathbf{x}\sim\mathbb{P}_{\text{out}}^{\text{semantic}}}(\mathbb{1}\{g_{\boldsymbol{\theta}}(\mathbf{x}) = \text{IN}\})$, where $\mathbb{1}\cdot$ is the indicator function. These metrics collectively assess ID generalization (ID-Acc), OOD generalization (OOD-Acc), and OOD detection performance (FPR), respectively.

**Novel Human-Assisted Learning Framework.** In addition to the initial labeled training set $\mathcal{S}_{\text{in}}$, a learning algorithm is also given a small budget of $B$ examples for human labeling. Before the annotation starts, we assume access to a set of wild examples $\mathcal{S}_{\text{wild}} = \{\mathbf{x}_i \sim \mathbb{P}_{\text{wild}}\}_{i=1}^N$, where their corresponding labels $\{y_i\}_{i=1}^N$ are unknown. For each example $\bar{\mathbf{x}} \in \mathcal{S}_{\text{wild}}$ chosen by the algorithm, a human assistant provides a ground truth label $\bar{y}$. Here, $\bar{y} = \text{OOD}$ if $\bar{\mathbf{x}}$ is semantic OOD. Otherwise, $\bar{y} \in [K]$ represents the class category of $\bar{\mathbf{x}}$ when it is ID or covariate OOD. We let $\mathcal{S}_{\text{human}} = \{(\bar{\mathbf{x}}_t, \bar{y}_t)\}_{t=1}^k$ denote the set of annotated wild examples. At last, neural networks $f_{\theta}$ and $g_{\theta}$ are trained on $\mathcal{S}_{\text{in}} \cup \mathcal{S}_{\text{human}}$. The objective of the learning algorithm is to choose and label $\mathcal{S}_{\text{human}}$ so that the performances of $f_{\theta}$ and $g_{\theta}$ are optimized (see learning goal above for metrics).

## 4 Methodology

In this section, we begin by identifying a good labeling region – a subset of wild examples that can significantly boost the OOD generalization and detection performances if labeled. We first present five straightforward yet novel baseline labeling regions for human-assisted OOD learning in Section 4.1. In Section 4.2, to address their limitations, we propose a significantly more effective labeling region, termed the *maximum disambiguation region*. This labeling region is an interval where the densities of the covariate and semantic OOD scores are roughly equal. We describe the details of the learning algorithm AHA (**A**daptive **H**uman-**A**ssisted OOD learning) in Section 4.3 to effectively identify and label examples in this region. Lastly in Section 4.4, we discuss the training objective we used to incorporate the human feedback for OOD learning.

### 4.1 Baseline Labeling Regions

Table 2: Practical labeling strategies, such as selecting the top-k most OOD examples and 95% true positive rate (TPR), display worse performance compared to AHA sampling. For experiments, we set a budget of $k = 500$. We train on CIFAR-10 as the ID dataset, using wild data with $\pi_c = 0.4$ (CIFAR-10-C) and $\pi_s = 0.3$ (Texture). The OOD score is measured using the energy score.

| | Labeling Regions | OOD Acc.↑ | ID Acc.↑ | FPR↓ | AUROC↑ | #ID | #Covariate OOD | #Semantic OOD |
|---|---|---|---|---|---|---|---|---|
| Oracle | Most Covariate OOD | 85.83 | 94.67 | 7.51 | 96.76 | 0 | 92 | 408 |
| | Least Semantic OOD | 80.84 | 94.88 | 27.10 | 86.00 | 325 | 127 | 48 |
| | Mixed Region | 83.12 | 94.84 | 9.68 | 95.88 | 177 | 79 | 244 |
| | **Maximum Disambiguation Region (ours)** | **88.93** | 94.54 | **4.81** | **98.17** | 14 | 237 | 249 |
| Practical | Top-$k$ Most Examples Region | 85.44 | 94.55 | 8.47 | 96.40 | 0 | 87 | 413 |
| | Near-Boundary Region | 87.55 | 94.77 | 9.50 | 95.73 | 67 | 263 | 170 |
| | **AHA (ours)** | **88.80** | 94.75 | **4.69** | **98.22** | 17 | 219 | 264 |

As shown in Figure 1, using a given OOD detection scoring function $g$ (we employ the energy score for our case study; see the appendix for a detailed description of different scoring function choices $g$), we can order the wild examples $\mathcal{S}_{\text{wild}}$ from the least to the most likely of being OOD.

Let $\mathcal{S}_{\text{wild}}^{\text{in}}$, $\mathcal{S}_{\text{wild}}^{\text{covariate}}$ and $\mathcal{S}_{\text{wild}}^{\text{semantic}}$ denote the sets of ID, covariate OOD and semantic OOD data in $\mathcal{S}_{\text{wild}}$ respectively. When collecting human feedback, the ideal outcome is to label examples that best separate ID and covariate OOD examples from the semantic OOD ones. This may be achieved by labeling *the highest score covariate OOD examples* and *the lowest score semantic OOD examples*. Let $\bar{\lambda}_{\text{covariate}} := \max_{\mathbf{x} \in \mathcal{S}_{\text{wild}}^{\text{covariate}}} g(\mathbf{x})$ and $\underline{\lambda}_{\text{semantic}} := \min_{\mathbf{x} \in \mathcal{S}_{\text{wild}}^{\text{semantic}}} g(\mathbf{x})$ denote the scores of the most covariate and the least semantic OOD examples. For analysis purposes, we propose the following three oracular labeling regions with a labeling budget of $k$:

- **Most covariate OOD**: Label the top-$k$ OOD score examples from $\{\mathbf{x} \in \mathcal{S}_{\text{wild}} : g(\mathbf{x}) \leq \bar{\lambda}_{\text{covariate}}\}$.
- **Least semantic OOD**: Label the bottom-$k$ OOD score examples from $\{\mathbf{x} \in \mathcal{S}_{\text{wild}} : g(\mathbf{x}) \geq \underline{\lambda}_{\text{semantic}}\}$.
- **Mixture of the two**: Allocate half of the budget $\frac{k}{2}$ to **most covariate OOD** and the remaining half $\frac{k}{2}$ to **least semantic OOD**, combining the two subsets.

In practice, since $\bar{\lambda}_{\text{covariate}}$ and $\underline{\lambda}_{\text{semantic}}$ are unknown, one may opt for the following two surrogate practical labeling regions:

- **Top-$k$ most OOD examples**: As a surrogate to the **most covariate OOD** labeling region, we label the top-$k$ OOD score examples from $\mathcal{S}_{\text{wild}}$ (see Figure 1 (a)).
- **Near-boundary examples**: As a surrogate to the **least semantic OOD** labeling region, we label $k$ examples closest to the $95\%$ TPR threshold from both sides (see Figure 1 (b)). We choose the threshold based on the labeled ID data $\mathcal{S}_{\text{in}}$, which captures a substantial fraction of ID examples (e.g., $95\%$), and is commonly defined as the ID vs OOD boundary in OOD detection literature.

**Limitations in OOD Learning Performance of Baseline Labeling Regions.** We conducted a case study on the five novel baseline labeling regions listed in Table 2. Although our proposed novel oracle **Most covariate OOD region** targets selecting covariate OOD with the highest scores, it performs poorly in wild settings. Most selected examples turn out to be semantic OOD near $\bar{\lambda}_{\text{covariate}}$, which does not aid in OOD generalization as expected. Similarly, the oracle **Least semantic OOD region** aims to identify semantic OOD examples with the lowest scores, and mostly ends up labeling ID and covariate OOD examples near $\underline{\lambda}_{\text{semantic}}$. The **Mixed range** achieves performance somewhere in between the two. We observe a similar phenomenon for the practical **Top-$k$ most examples region** and **Near-boundary region**. The above labeling regions are not as effective one might hope. This is primarily due to the dominant number of the other types of data around the most covariate OOD and least semantic OOD examples, which are not informative. This motivates us to label examples where the density of the two types of OOD examples roughly equalizes—the maximum disambiguation region. Empirically, as shown in Table 2, we observe that labeling within this region can significantly improve overall performance in both oracle and practical settings.

### 4.2 Maximum Disambiguation Region

In this section, we formally introduce the maximum disambiguation region (see Figure 1(c)), centered around the *maximum ambiguity threshold*. While we hope to find the threshold where the densities of semantic and covariate OOD examples equalize, it is impossible to distinguish between covariate OOD examples from ID examples based on human labels. Therefore, we formally define the threshold as the OOD score where the weighted density of semantic OOD examples is equal to that of covariate OOD and ID examples combined.

Concretely, given the OOD scoring function $g$, we let $p_{\text{covariate}}(\mu)$ be the probability density of $g(\mathbf{x})$ when $\mathbf{x}$ is drawn from the covariate OOD distribution. That is, $\int_0^\mu p_{\text{covariate}}(\nu)d\nu$ is the probability that an $\mathbf{x}$ drawn from the covariate OOD distribution has a score less than or equal to $\mu$. Similarly, we define $p_{\text{in}}$ and $p_{\text{semantic}}$ as the probability densities of $g(\mathbf{x})$ when $\mathbf{x}$ is drawn from ID and semantic OOD distributions respectively. Recall $\pi_c$ and $\pi_s$ are the prior probabilities of x coming from the covariate and semantic OOD distributions, we define the maximum ambiguity threshold as follows.

**Definition 1** (Maximum Ambiguity Threshold)**.** *Given the OOD scoring for all wild data points, we define the maximum ambiguity threshold as the CDF of the two categories of examples is maximized:*

$$\lambda_* = \arg\max_{\mu \in \mathbb{R}} \int_0^\mu ((1 - \pi_c - \pi_s)p_{in}(\nu) + \pi_c p_{covariate}(\nu)) - \pi_s p_{semantic}(\nu)d\nu. \tag{1}$$

*Ties are broken by choosing the $\mu$ value closest to the median of the OOD scores of the wild examples.* Note that under benign continuity assumptions, we necessarily have $(1 - \pi_c - \pi_s) \cdot p_{\text{in}}(\lambda^*) + \pi_c p_{\text{covariate}}(\lambda^*) = \pi_s p_{\text{semantic}}(\lambda^*)$, where the weighted densities of the two distributions equalize.

Through a different lens, the threshold $\lambda_*$ also corresponds to where the current OOD detector is most *uncertain* about its prediction. In fact, when we label around this threshold, we make the maximum number of corrections to the OOD detector's predictions, correcting at least half of the examples to their appropriate categories.

**Reduction to noisy binary search.** At the essence, the above is a noisy binary search problem. When labeling an examples $\mathbf{x}$ with OOD score $\nu = g(\mathbf{x})$, the outcome is a Bernoulli-like random variable. Specifically, one observes a class label $y \in [K]$ with probability $p_{\text{in}}(\nu) + p_{\text{covariate}}(\nu)$, and an $y = \texttt{OOD}$ label with probability $p_{\text{semantic}}(\nu)$. When given a labeled set $\mathcal{S} = \{(\bar{\mathbf{x}}_i, \bar{y}_i)\}_{i \in [n]}$ of size $n$, by finite sample approximation, equation (1) can be further derived as

$$\max_{\mu \in \mathbb{R}} \int_0^\mu ((1 - \pi_c - \pi_s)p_{\text{in}}(\nu) + \pi_c p_{\text{covariate}}(\nu)) - \pi_s p_{\text{semantic}}(\nu)d\nu \tag{2}$$

$$\approx \max_{\mu \in \mathbb{R}} |\{y_i \neq \texttt{OOD} : (\mathbf{x}_i, y_i) \in \mathcal{S}, g(\mathbf{x}_i) \leq \mu\}| - |\{y_i = \texttt{OOD} : (\mathbf{x}_i, y_i) \in \mathcal{S}, g(\mathbf{x}_i) \leq \mu\}|. \tag{3}$$

### 4.3 Algorithm

Our algorithm AHA consists of two main steps: (1) We propose identifying the maximum ambiguity threshold by leveraging an off-the-shelf adaptive labeling algorithm [56]. This threshold is determined by equation 2, where the cumulative number of ID and covariate OOD examples most dominate that of semantic OOD examples. (2) We then annotate an equal number of examples on both sides of this identified maximum ambiguity threshold, establishing the maximum disambiguation region.

Specifically, as shown in Algorithm 1, AHA starts by initializing an empty set for labeled examples and a broad confidence interval for the maximum ambiguity threshold. During the first phase, the algorithm iteratively and adaptively labels more examples. Over the annotation period, our algorithm maintains a confidence interval $[\underline{\mu}, \bar{\mu}]$ with high probability, ensuring that the maximum ambiguity threshold $\lambda_* \in [\underline{\mu}, \bar{\mu}]$ lies within this interval with high probability. During each iteration of the first phase, we uniformly at random label an example within this confidence interval. Upon obtaining the label, we update the confidence interval using a subprocedure called **ConfUpdate**. This subprocedure shrinks the interval based on the labeled examples, ensuring it converges to an accurate threshold over time with statistical guarantees. The detailed implementations of **ConfUpdate** and its theoretical foundations are discussed in Appendix A and [56] respectively. Overall, we spend half of our labeling budget during the first phase. During the second phase, we then spend the remaining half of the budget labeling examples around the identified threshold. Finally, the classifier and the OOD detector are trained on the combined set of initially labeled and newly annotated examples.

---

**Algorithm 1** AHA: Adaptive Human Assisted labeling for OOD learning

---

**Input:** OOD detector $g$ trained on $\mathcal{S}_{\text{in}}$, wild set of examples $\mathcal{S}_{\text{wild}} = \{\mathbf{x}_i\}_{i=1}^N$, budget $k$
**Initialize:** $\mathcal{S}_{\text{human}} \leftarrow \{\}$, confidence interval $\underline{\mu}, \bar{\mu} \leftarrow -\infty, \infty$
**Spend half budget searching for maximum ambiguity threshold**

**for** $t = 1, ..., \frac{k}{2}$ **do**
    Sample $\bar{\mathbf{x}}_t$ uniformly at random from $\{\mathbf{x} \in \mathcal{S}_{\text{wild}} \backslash \mathcal{S}_{\text{human}} : \underline{\mu} \leq g(\mathbf{x}) \leq \bar{\mu}\}$
    Ask human for label on $\bar{\mathbf{x}}_t$, observe $\bar{y}_t$, and insert the example $(\bar{\mathbf{x}}_t, \bar{y}_t)$ into $\mathcal{S}_{\text{human}}$
    Update confidence interval $\underline{\mu}, \bar{\mu} \leftarrow$ **ConfUpdate**$(\mathcal{S}_{\text{human}}, \mathcal{S}_{\text{wild}}, g, \underline{\mu}, \bar{\mu})$
**end for**
**Spend half budget labeling around identified threshold**
Compute $\hat{\mu}$ as an arbitrary solution that reaches the maximum in equation (2)
Label examples **Top**$(\frac{k}{4}, \{\mathbf{x} \in \mathcal{S}_{\text{wild}} \backslash \mathcal{S}_{\text{human}} : g(\mathbf{x}) \leq \hat{\mu}\}; g)$ and **Bottom**$(\frac{k}{4}, \{\mathbf{x} \in \mathcal{S}_{\text{wild}} \backslash \mathcal{S}_{\text{human}} : g(\mathbf{x}) > \hat{\mu}\}; g)$ and insert them in $\mathcal{S}_{\text{human}}$
**Return:** New classifier $f_{\mathbf{w}}$ and OOD detector $g_\theta$ trained on $\mathcal{S}_{\text{in}} \cup \mathcal{S}_{\text{human}}$ based on Section 4.4

---

### 4.4 Learning Objective

Let $\mathcal{S}_{\text{human}}^c$ denote the set of annotated covariate examples, and $\mathcal{S}_{\text{human}}^s$ represent the set of annotated semantic examples from wild data. Our learning framework jointly optimizes two objectives: (1) multi-class classification of examples from $\mathcal{S}_{\text{in}}$ and covariate OOD $\mathcal{S}_{\text{human}}^c$, and (2) a binary OOD

detector separating data between $\mathcal{S}_{\text{in}}$ and semantic OOD $\mathcal{S}_{\text{human}}^{\text{s}}$. The risk formulation is defined as:

$$\mathbf{w}, \boldsymbol{\theta} = \arg\min[\underbrace{R_{\mathcal{S}_{\text{in}}, \mathcal{S}_{\text{human}}^{\text{c}}}(f_{\mathbf{w}})}_{\text{Multi-class classifier}} + \alpha \cdot \underbrace{R_{\mathcal{S}_{\text{in}}, \mathcal{S}_{\text{human}}^{\text{s}}}(g_{\boldsymbol{\theta}})}_{\text{OOD detector}}], \tag{4}$$

where $\alpha$ is the weighting factor. The first term is optimized using standard cross-entropy loss, and the second term is aimed at explicitly optimizing the level-set based on the model output (threshold at 0):

$$\begin{aligned} R_{\mathcal{S}_{\text{in}}, \mathcal{S}_{\text{human}}^{\text{s}}}(g_{\boldsymbol{\theta}}) &= R_{\mathcal{S}_{\text{in}}}^{+}(g_{\boldsymbol{\theta}}) + R_{\mathcal{S}_{\text{human}}^{\text{s}}}^{-}(g_{\boldsymbol{\theta}}) \\ &= \mathbb{E}_{\mathbf{x} \in \mathcal{S}_{\text{in}}} \ \mathbb{1}\{g_{\boldsymbol{\theta}}(\mathbf{x}) \leq 0\} + \mathbb{E}_{\tilde{\mathbf{x}} \in \mathcal{S}_{\text{human}}^{\text{s}}} \ \mathbb{1}\{g_{\boldsymbol{\theta}}(\tilde{\mathbf{x}}) > 0\}. \end{aligned} \tag{5}$$

We replace the $0/1$ loss with the binary sigmoid loss as a smooth approximation to the $0/1$ loss.

## 5 Experiments

In this section, we comprehensively evaluate the efficacy of the AHA for OOD generalization and detection. First, we describe the experimental setup in Section 5.1. In Section 5.2, we present the main results and discussion on OOD generalization and detection. Then, we provide ablation studies to further understand the human-assisted OOD learning framework (Section 5.3).

### 5.1 Experiment Setup

**Datasets and evaluation metrics.** Following the benchmark in literature of [6], we use the CIFAR-10 [60] as $\mathbb{P}_{\text{in}}$ and CIFAR-10-C [45] with Gaussian additive noise as the $\mathbb{P}_{\text{out}}^{\text{covariate}}$ for our main experiments. We also provide ablations on other types of covariate OOD data in the Appendix J. For semantic OOD data ($\mathbb{P}_{\text{out}}^{\text{semantic}}$), we utilize natural image datasets including SVHN [72], Textures [19], Places365 [113], LSUN-Crop [103], and LSUN-Resize [103]. Additionally, we provide results on the PACS dataset [64] from DomainBed. Large-scale results on the ImageNet dataset can be found in Appendix F A detailed description of the datasets is presented in Appendix D. To compile the wild data, we divide the ID set into 50% labeled as ID (in-distribution) and 50% unlabeled. We then mix unlabeled ID, covariate OOD, and semantic OOD data for our experiments. To simulate the wild distribution $\mathbb{P}_{\text{wild}}$, we adopt the same mixture ratio used in the benchmark of SCONE [6], where $\pi_c = 0.5$ and $\pi_s = 0.1$. We also evaluate different wild mixture rates in Section 5.3. For evaluation, we use the collection of metrics defined in Section 3. The threshold for the OOD detector is selected based on the ID data when 95% of ID test data points are correctly classified as ID.

**Experimental details.** For CIFAR experiments, we adopt a Wide ResNet [104] with 40 layers and a widen factor of 2. For optimization, we use stochastic gradient descent with Nesterov momentum [27], including a weight decay of 0.0005 and a momentum of 0.09. The batch size is set to 128, and the initial learning rate is 0.1, with cosine learning rate decay. The model is initialized with a pre-trained network on CIFAR-10 and trained for 100 epochs using our objective from Equation 4, with $\alpha = 10$. We set a default labeling budget $k$ of 1000 for the benchmarking results and provide an

Table 3: **Main results**: comparison with competitive OOD generalization and OOD detection methods on CIFAR-10. *Since all the OOD detection methods use the same model trained with the CE loss on $\mathbb{P}_{\text{in}}$, they display the same ID and OOD accuracy on CIFAR-10-C. We report the average and standard error ($\pm x$) of our method based on three independent runs.

| Method | SVHN $\mathbb{P}_{\text{out}}^{\text{semantic}}$, CIFAR-10-C $\mathbb{P}_{\text{out}}^{\text{covariate}}$ | | | | LSUN-C $\mathbb{P}_{\text{out}}^{\text{semantic}}$, CIFAR-10-C $\mathbb{P}_{\text{out}}^{\text{covariate}}$ | | | | Texture $\mathbb{P}_{\text{out}}^{\text{semantic}}$, CIFAR-10-C $\mathbb{P}_{\text{out}}^{\text{covariate}}$ | | | |
|---|---|---|---|---|---|---|---|---|---|---|---|---|
| | OOD Acc.↑ | ID Acc.↑ | FPR↓ | AUROC↑ | OOD Acc.↑ | ID Acc.↑ | FPR↓ | AUROC↑ | OOD Acc.↑ | ID Acc.↑ | FPR↓ | AUROC↑ |
| *OOD detection* | | | | | | | | | | | | |
| MSP | 75.05* | 94.84* | 48.49 | 91.89 | 75.05 | 94.84 | 30.80 | 95.65 | 75.05 | 94.84 | 59.28 | 88.50 |
| ODIN | 75.05 | 94.84 | 33.35 | 91.96 | 75.05 | 94.84 | 15.52 | 97.04 | 75.05 | 94.84 | 49.12 | 84.97 |
| Energy | 75.05 | 94.84 | 35.59 | 90.96 | 75.05 | 94.84 | 8.26 | 98.35 | 75.05 | 94.84 | 52.79 | 85.22 |
| Mahalanobis | 75.05 | 94.84 | 12.89 | 97.62 | 75.05 | 94.84 | 39.22 | 94.15 | 75.05 | 94.84 | 15.00 | 97.33 |
| ViM | 75.05 | 94.84 | 21.95 | 95.48 | 75.05 | 94.84 | 5.90 | 98.82 | 75.05 | 94.84 | 29.35 | 93.70 |
| KNN | 75.05 | 94.84 | 28.92 | 95.71 | 75.05 | 94.84 | 28.08 | 95.33 | 75.05 | 94.84 | 39.50 | 92.73 |
| ASH | 75.05 | 94.84 | 40.76 | 90.16 | 75.05 | 94.84 | 2.39 | 99.35 | 75.05 | 94.84 | 53.37 | 85.63 |
| *OOD generalization* | | | | | | | | | | | | |
| ERM | 75.05 | 94.84 | 35.59 | 90.96 | 75.05 | 94.84 | 8.26 | 98.35 | 75.05 | 94.84 | 52.79 | 85.22 |
| Mixup | 79.17 | 93.30 | 97.33 | 18.78 | 79.17 | 93.30 | 52.10 | 76.66 | 79.17 | 93.30 | 58.24 | 75.70 |
| IRM | 77.92 | 90.85 | 63.65 | 90.70 | 77.92 | 90.85 | 36.67 | 94.22 | 77.92 | 90.85 | 59.42 | 87.81 |
| VREx | 76.90 | 91.35 | 55.92 | 91.22 | 76.90 | 91.35 | 51.50 | 91.56 | 76.90 | 91.35 | 65.45 | 85.46 |
| EQRM | 75.71 | 92.93 | 51.86 | 90.92 | 75.71 | 92.93 | 21.53 | 96.49 | 75.71 | 92.93 | 57.18 | 89.11 |
| SharpDRO | 79.03 | 94.91 | 21.24 | 96.14 | 79.03 | 94.91 | 5.67 | 98.71 | 79.03 | 94.91 | 42.94 | 89.99 |
| *Learning w. $\mathbb{P}_{\text{wild}}$* | | | | | | | | | | | | |
| OE | 37.61 | 94.68 | 0.84 | 99.80 | 41.37 | 93.99 | 3.07 | 99.26 | 44.71 | 92.84 | 29.36 | 93.93 |
| Energy (w. outlier) | 20.74 | 90.22 | 0.86 | 99.81 | 32.55 | 92.97 | 2.33 | 99.93 | 49.34 | 94.68 | 16.42 | 96.46 |
| WOODS | 52.76 | 94.86 | 2.11 | 99.52 | 76.90 | 95.02 | 1.80 | 99.56 | 83.14 | 94.49 | 39.10 | 90.45 |
| SCONE | 84.69 | 94.65 | 0.08 | 99.99 | 84.58 | 93.73 | 10.23 | 98.02 | 85.56 | 93.97 | 37.15 | 90.91 |
| **AHA (Ours)** | **89.01**$_{\pm 0.01}$ | 94.67$_{\pm 0.00}$ | **0.08**$_{\pm 0.00}$ | **99.99**$_{\pm 0.00}$ | **90.69**$_{\pm 0.11}$ | 94.45$_{\pm 0.07}$ | **0.02**$_{\pm 0.01}$ | **99.98**$_{\pm 0.01}$ | **90.51**$_{\pm 0.06}$ | 94.54$_{\pm 0.02}$ | **5.63**$_{\pm 0.20}$ | **97.95**$_{\pm 0.14}$ |

analysis of different labeling budgets 100, 500, 1000, 2000 in Section 5.3. In our experiment, the output of $g_\theta$ is utilized as the score for OOD detection.

## 5.2 Main Results and Discussion

**Results on benchmark for both OOD generalization and detection.** Table 3 provides a comparative analysis of various OOD generalization and detection methods on the CIFAR benchmark, evaluating their performance across different semantic OOD datasets including SVHN, LSUN-C, and Textures. AHA shows significant improvements for both OOD generalization and OOD detection tasks, suggesting a robust method for handling OOD scenarios.

Specifically, we compare AHA with three groups of methods: (1) methods developed for OOD generalization, including IRM [2], GroupDRO [81], Mixup [106], VREx [61], EQRM [29], and the more recent SharpDRO [51]; (2) methods tailored for OOD detection, including MSP [46], ODIN [67], Energy [68], Mahalanobis [62], ViM [98], KNN [91], and the more recent ASH [25]; and (3) methods that are trained with unlabeled data from the wild, including Outlier Exposure [47], Energy-based Regularized Learning [68], WOODS [55], and SCONE [6].

We highlight some key observations: (1) AHA achieves superior performance compared to specifically designed OOD generalization baselines. These baselines struggle to distinguish between ID data and semantic OOD data, leading to poor OOD detection performance. Additionally, our method selects the optimal region and involves human labeling to retrain the model using the selected examples, thus leading to better generalization performance compared to other OOD generalization baselines. (2) Our approach achieves superior performance compared to OOD detection baselines. Methods specifically designed for OOD detection, which aim to identify and separate semantic OOD, show suboptimal OOD accuracy. This demonstrates that existing OOD detection baselines struggle with covariate distribution shifts. (3) Compared with strong baselines trained with wild data, AHA consistently outperforms existing learning with wild data baselines. Specifically, our approach surpasses the current state-of-the-art (SOTA) method, SCONE, by 31.52% in terms of FPR95 on the Texture OOD dataset and simultaneously improves the OOD accuracy by 4.95%. This demonstrates the robust effectiveness of our method for both OOD generalization and detection tasks.

**Additional results on PACS.** Table 4 presents our results on the PACS dataset [64] from DomainBed [40]. We compare AHA against various common OOD generalization baselines, including IRM [2], DANN [37], CDANN [66], GroupDRO [81], MTL [14], I-Mixup [101], MMD [65], VREx [61], MLDG [63], ARM [111], RSC [50], Mixstyle [116], ERM [95], CORAL [89], SagNet [71], SelfReg [57], GVRT [69], VNE [58], and the most recent baseline HYPO [7]. Our method achieves an average accuracy of 92.7%, outperforming these OOD generalization baselines.

Table 4: Comparison with domain generalization methods on the PACS benchmark. We followed the same leave-one-domain-out validation experimental protocol as in [64]. All methods are trained on ResNet-50. The model selection is based on a training domain validation set.

| Algorithm | Art | Cartoon | Photo | Sketch | Average |
|---|---|---|---|---|---|
| IRM [2] | 84.8 | 76.4 | 96.7 | 76.1 | 83.5 |
| DANN [37] | 86.4 | 77.4 | 97.3 | 73.5 | 83.7 |
| CDANN [66] | 84.6 | 75.5 | 96.8 | 73.5 | 82.6 |
| GroupDRO [82] | 83.5 | 79.1 | 96.7 | 78.3 | 84.4 |
| MTL [14] | 87.5 | 77.1 | 96.4 | 77.3 | 84.6 |
| I-Mixup [101] | 86.1 | 78.9 | 97.6 | 75.8 | 84.6 |
| MMD [65] | 86.1 | 79.4 | 96.6 | 76.5 | 84.7 |
| VREx [61] | 86.0 | 79.1 | 96.9 | 77.7 | 84.9 |
| MLDG [63] | 85.5 | 80.1 | 97.4 | 76.6 | 84.9 |
| ARM [111] | 86.8 | 76.8 | 97.4 | 79.3 | 85.1 |
| RSC [50] | 85.4 | 79.7 | 97.6 | 78.2 | 85.2 |
| Mixstyle [116] | 86.8 | 79.0 | 96.6 | 78.5 | 85.2 |
| ERM [95] | 84.7 | 80.8 | 97.2 | 79.3 | 85.5 |
| CORAL [89] | 88.3 | 80.0 | 97.5 | 78.8 | 86.2 |
| SagNet [71] | 87.4 | 80.7 | 97.1 | 80.0 | 86.3 |
| SelfReg [57] | 87.9 | 79.4 | 96.8 | 78.3 | 85.6 |
| GVRT [69] | 87.9 | 78.4 | 98.2 | 75.7 | 85.1 |
| VNE [58] | 88.6 | 79.9 | 96.7 | 82.3 | 86.9 |
| HYPO [7] | 90.5 | 84.6 | 97.7 | 83.2 | 89.0 |
| **AHA (Ours)** | **92.6** | **93.5** | **98.7** | **86.1** | **92.7** |

Table 5: Impact of sampling scores with our selection strategy. We use budget $k = 1000$ for all methods. We train on CIFAR-10 as ID, using wild data with $\pi_c = 0.5$ (CIFAR-10-C) and $\pi_s = 0.1$ (Texture).

| Sampling score | OOD Acc.↑ | ID Acc.↑ | FPR↓ | AUROC↑ |
|---|---|---|---|---|
| Random | 89.22 | 94.84 | 9.45 | 95.41 |
| Least confidence | 90.08 | 94.40 | 5.29 | 97.94 |
| Entropy | 89.99 | 94.50 | 5.35 | 97.75 |
| Margin | 90.10 | 94.55 | **4.15** | **98.53** |
| Energy score | 89.58 | 94.73 | 6.37 | 97.26 |
| Gradient-based | **90.51** | 94.54 | 5.63 | 97.95 |

Table 6: Ablation on labeling budget $k$. We train on CIFAR-10 as ID, using wild data with $\pi_c = 0.4$ (CIFAR-10-C) and $\pi_s = 0.3$ (Texture).

| Budget | Method | OOD Acc.↑ | ID Acc.↑ | FPR↓ | AUROC↑ |
|---|---|---|---|---|---|
| 100 | **Top-k** | 79.77 | 94.89 | 17.55 | 91.98 |
| | **AHA (Ours)** | **85.07** | 94.93 | **14.78** | **92.79** |
| 500 | **Top-k** | 85.44 | 94.55 | 8.47 | 96.40 |
| | **AHA (Ours)** | **88.80** | 94.75 | **4.69** | **98.22** |
| 1000 | **Top-k** | 88.32 | 94.51 | 5.41 | 97.62 |
| | **AHA (Ours)** | **89.46** | 94.50 | **3.19** | **98.83** |
| 2000 | **Top-k** | 89.87 | 94.47 | 2.64 | 99.05 |
| | **AHA (Ours)** | **90.46** | 94.41 | **2.04** | **99.17** |

## 5.3 Ablation Studies

**Effect of different scores.** Different OOD scores play a crucial role in identifying various distributions and impacting the selection process. To evaluate the effectiveness of different OOD scores within our framework, we conducted an ablation study (see Table 5). Detailed descriptions of the different OOD scores can be found in Appendix C. The scores include least-confidence [97, 46], entropy [97], margin [80], energy score [68], gradient-based [26]. We also compared our approach with random sampling, which serves as a straightforward baseline method involving the random selection of $k$ examples to query. We observe that AHA consistently achieves superior performance when combined with various sampling scores for OOD generalization and detection, and it consistently outperforms the random sampling baseline. The gradient-based score demonstrates the best overall performance in terms of OOD accuracy and FPR. This also shows that AHA can be easily integrated with existing sampling scores.

**Effect on different labeling budgets $k$.** In Table 6, we provide ablations on different labeling budgets $k$ from 100, 500, 1000, 2000. We observe that both OOD generalization and detection performance improve with an increasing labeling budget. For instance, our method's OOD accuracy increased from 79.77% to 90.46% when the budget increased from 100 to 2000. Simultaneously, the TPR decreased from 17.55% to 2.04%, which also indicates a significant improvement in OOD detection performance. Moreover, AHA consistently outperforms the practical top-$k$ OOD example sampling strategy across different labeling budgets.

## 5.4 Qualitative Analysis

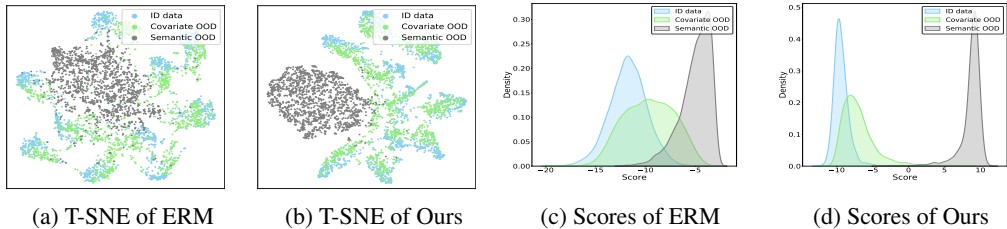

| (a) T-SNE of ERM | (b) T-SNE of Ours | (c) Scores of ERM | (d) Scores of Ours |

Figure 2: (a)-(b): T-SNE visualization of the image embeddings for ERM vs. AHA (ours). (c)-(d) Score distributions for ERM vs. AHA (ours). Different colors represent the different types of test data: CIFAR-10 as $\mathbb{P}_{in}$ (blue), CIFAR-10-C as $\mathbb{P}_{out}^{covariate}$ (green), and Textures as $\mathbb{P}_{out}^{semantic}$ (gray).

**Visualization of feature embeddings.** Figure 2 (a) and (b) present feature embedding visualizations using t-SNE [93] on the test data. The blue points represent the ID test data (CIFAR-10), green points represent OOD test examples from CIFAR-10-C, and gray points are from the Texture dataset. We observe that (1) the embeddings of the ID data $\mathbb{P}_{in}$ (CIFAR-10) and the covariate shift OOD data $\mathbb{P}_{out}^{covariate}$ (CIFAR-10-C) are more closely aligned, and (2) the embeddings of the semantic shift OOD data $\mathbb{P}_{out}^{covariate}$ (Texture) are better separated from the ID and covariate shift OOD data using our method. This contributes to enhanced OOD generalization and OOD detection performance.

**Visualization of OOD score distributions.** Figure 2 (c) and (d) visualize the score distributions using kernel density estimation (KDE) for the baseline and our method. The OOD score distributions between the ID data ($\mathbb{P}_{in}$) and the semantic OOD data ($\mathbb{P}_{out}^{semantic}$) are more separated using our method. This separation represents an improvement in OOD detection performance, demonstrating the effectiveness of AHA in identifying semantic OOD data.

## 6 Conclusion

In this study, we introduce the first human-assisted framework designed to simultaneously address OOD generalization and OOD detection by leveraging wild data. We propose a novel labeling strategy that selects the maximum disambiguation region, strategically utilizing human labels to maximize model performance amid covariate and semantic shifts. Extensive experiments demonstrate that AHA effectively enhances both OOD generalization and detection performance. This research establishes a solid foundation for further advancements in OOD learning within dynamic environments characterized by heterogeneous data shifts.

## Acknowledgement

This work has been supported in part by NSF Award 2112471.

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

# AHA: Human-Assisted Out-of-Distribution Generalization and Detection (Appendix)

## A    ConfUpdate: Shrinking Confidence Interval for Labeling

Given the current interval $[\underline{\mu}, \bar{\mu}]$, the OOD scoring function $g$ and the wild dataset $\mathcal{S}_{\text{wild}}$, we can denote $\mathcal{S}_{[\underline{\mu}, \bar{\mu}]}$ as the set of examples $\mathcal{S}_{[\underline{\mu}, \bar{\mu}]} := \{\mathbf{x} \in \mathcal{S}_{\text{wild}} : \underline{\mu} \leq g(x) \leq \bar{\mu}\}$. The goal of **ConfUpdate** is to shrink the two ends of this interval of examples so that its size shrink by a factor of $c := \sqrt[k/2]{|\mathcal{S}_{\text{wild}}|}$. This way, after labeling $\frac{k}{2}$ examples, the confidence interval would only contain a single example.

We let the examples $\mathbf{x}_{(1)}, ..., \mathbf{x}_{(m)}$ denote ordered list of examples in $\mathcal{S}_{[\underline{\mu}, \bar{\mu}]}$ based on the OOD scoring function $g$. Note $m = |\mathcal{S}_{[\underline{\mu}, \bar{\mu}]}|$, the confidence interval is shrunk by finding $I, J \in [m]$ where $I < J$ such that:

$$I, J = \underset{i, j: j - i = \frac{1}{c}m}{\arg\min} \max\{\widehat{\mathcal{L}}(i), \widehat{\mathcal{L}}(j)\} \quad \text{where}$$

$$\widehat{\mathcal{L}}(s) = \sum_{r \leq s: \mathbf{x}_{(r)} \in \mathcal{S}_{\text{human}}} \mathbf{1}\{y_{(r)} \neq \texttt{OOD}\} - \sum_{r \leq s: \mathbf{x}_{(r)} \in \mathcal{S}_{\text{human}}} \mathbf{1}\{y_{(r)} = \texttt{OOD}\}.$$

Here, $\mathcal{S}_{\text{human}}$ is the labeled set of examples from input to the algorithm. Each example in this set, therefore has a corresponding label $y$. The loss $\widehat{\mathcal{L}}(s)$ is an empirical estimate of the loss in equation 2. Intuitively, this shrinking procedure is choosing the fixed-size subset interval that will result in the lowest empirical loss estimate based on current labeled examples.

Finally, we return $g(\mathbf{x}_I), g(\mathbf{x}_J)$ as the new confidence interval for $\underline{\mu}, \bar{\mu}$.

## B    Main Notations and Their Descriptions

Table 7: Main notations and their descriptions.

| Notation | Description |
|---|---|
| Spaces | |
| $\mathcal{X}, \mathcal{Y}$ | the input space and the label space. |
| Distributions | |
| $\mathbb{P}_{\text{wild}}, \mathbb{P}_{\text{in}}$ | data distribution for wild data and ID data. |
| $\mathbb{P}_{\text{out}}^{\text{covariate}}$ | data distribution for covariate-shifted OOD data. |
| $\mathbb{P}_{\text{out}}^{\text{semantic}}$ | data distribution for semantic-shifted OOD data. |
| $\mathbb{P}_{\mathcal{X}\mathcal{Y}}$ | the joint data distribution for ID data. |
| Data and Models | |
| $\mathcal{S}_{\text{in}}, \mathcal{S}_{\text{wild}}$ | labeled ID data and unlabeled wild data |
| $\mathcal{S}_{\text{human}}$ | labeled data |
| $\mathcal{S}_{\text{human}}^{\text{s}}, \mathcal{S}_{\text{human}}^{\text{c}}$ | semantic and covariate OOD in the labeled data $\mathcal{S}_{\text{human}}$ |
| $f_{\mathbf{w}}$ and $g_{\boldsymbol{\theta}}$ | predictor on labeled in-distribution and binary predictor for OOD detection |
| $y$ | label for ID classification |
| $\widehat{y}_{\mathbf{x}}$ | Predicted one-hot label for input $\mathbf{x}$ |
| $n, m, k$ | size of $\mathcal{S}^{\text{in}}$, size of $\mathcal{S}_{\text{wild}}$, labeling budget |
| Algorithm and Labeling Region | |
| $\bar{\lambda}_{\text{covariate}}$ | highest OOD score of covariate OOD examples |
| $\underline{\lambda}_{\text{semantic}}$ | lowest OOD score of semantic OOD examples |
| $\lambda_*$ | maximum disambiguity threshold |
| $\underline{\mu}, \bar{\mu}$ | high probability confidence set of possible location of the maximum ambiguity threshold |

# C  Description of Different OOD Scores functions $g$

In this section, we provide a detailed description of the different scoring function choices for $g$, which have been shown to work well for detecting semantic OOD images. Our proposed good labeling region method is orthogonal to post hoc OOD scores, allowing it to be integrated with various OOD score functions.

**MSP** [46, 97] is a simple baseline OOD score that uses probabilities from softmax distributions. This score focuses on instances where the model's predictions are least certain.

**Margin** [80] refers to the multiclass margin value for each point, specifically calculating the discrepancy between the posterior probabilities of the two most likely labels. Most OOD examples have closely matched posterior probabilities, indicating a minimal difference between them.

**Entropy score** [46, 97] quantifies how evenly spread the model's probabilistic predictions are among all $K$ classes. We calculate the entropy within the predictive class probability distribution of each example.

**Energy score** [68] identifies data points based on an energy score, which is theoretically aligned with the probability density of the inputs. This score evaluates the likelihood of each input belonging to the known distribution, providing a robust measure for distinguishing between ID and OOD examples.

**Gradient-based score** [26] leverages the gradients of the loss function to differentiate between ID and OOD data. This gradient-based filtering score provides a robust mechanism for identifying OOD in the wild by leveraging the inherent differences in gradient behaviors between ID and OOD data.

# D  Detailed Description of Datasets

In this section, we provide a detailed description of the datasets used in this work.

**CIFAR-10** [60] includes $60,000$ color images in 10 different classes, with 6,000 images per class. This is a widely used benchmark in machine learning and computer vision. The training set consists of $50,000$ images, while the test set comprises $10,000$ images.

**CIFAR-10-C** is generated based on the previous leterature [45]. The corruption types include Gaussian noise, defocus blur, glass blur, impulse noise, shot noise, snow, zoom blur, brightness, elastic transform, contrast, fog, forest, Gaussian blur, jpeg, motion blur, pixelate, saturate, spatter, and speckle noise.

**ImageNet-100** is a dataset composed of 100 categories randomly sampled from the ImageNet-1K dataset [24]. The classes included in ImageNet-100 are as follows:n01498041, n01514859, n01582220, n01608432, n01616318, n01687978, n01776313, n01806567, n01833805, n01882714, n01910747, n01944390, n01985128, n02007558, n02071294, n02085620, n02114855, n02123045, n02128385, n02129165, n02129604, n02165456, n02190166, n02219486, n02226429, n02279972, n02317335, n02326432, n02342885, n02363005, n02391049, n02395406, n02403003, n02422699, n02442845, n02444819, n02480855, n02510455, n02640242, n02672831, n02687172, n02701002, n02730930, n02769748, n02782093, n02787622, n02793495, n02799071, n02802426, n02814860, n02840245, n02906734, n02948072, n02980441, n02999410, n03014705, n03028079, n03032252, n03125729, n03160309, n03179701, n03220513, n03249569, n03291819, n03384352, n03388043, n03450230, n03481172, n03594734, n03594945, n03627232, n03642806, n03649909, n03661043, n03676483, n03724870, n03733281, n03759954, n03761084, n03773504, n03804744, n03916031, n03938244, n04004767, n04026417, n04090263, n04133789, n04153751, n04296562, n04330267, n04371774, n04404412, n04465501, n04485082, n04507155, n04536866, n04579432, n04606251, n07714990, n07745940.

**LSUN** [103] is a large image dataset with categories labeled using deep learning with humans in the loop. LSUN-C is a cropped version of LSUN, and LSUN-R is a resized version of the LSUN.

**Textures** [19] contains images of patterns and textures. The subset we use for the OOD detection task has no overlapping categories with the CIFAR dataset.

**SVHN** [72] is a natural image dataset containing house numbers from street-level photos, cropped from Street View images. This dataset includes 10 classes, with $73,257$ training examples and $26,032$ testing examples.

Table 8: Additional results. Comparison with competitive OOD detection and OOD generalization methods on CIFAR-10. For experiments using $\mathbb{P}_{\text{wild}}$, we use $\pi_s = 0.5$, $\pi_c = 0.1$. For each semantic OOD dataset, we create corresponding wild mixture distribution $\mathbb{P}_{\text{wild}} := (1 - \pi_s - \pi_c)\mathbb{P}_{\text{in}} + \pi_s \mathbb{P}_{\text{out}}^{\text{semantic}} + \pi_c \mathbb{P}_{\text{out}}^{\text{covariate}}$ for training. We report the average and standard error ($\pm x$) of our method based on three independent runs.

| Model | Places365 $\mathbb{P}_{\text{out}}^{\text{semantic}}$, CIFAR-10-C $\mathbb{P}_{\text{out}}^{\text{covariate}}$ | | | | LSUN-R $\mathbb{P}_{\text{out}}^{\text{semantic}}$, CIFAR-10-C $\mathbb{P}_{\text{out}}^{\text{covariate}}$ | | | |
| | OOD Acc.↑ | ID Acc.↑ | FPR↓ | AUROC↑ | OOD Acc.↑ | ID Acc.↑ | FPR↓ | AUROC↑ |
|---|---|---|---|---|---|---|---|---|
| *OOD detection* | | | | | | | | |
| **MSP** | 75.05 | 94.84 | 57.40 | 84.49 | 75.05 | 94.84 | 52.15 | 91.37 |
| **ODIN** | 75.05 | 94.84 | 57.40 | 84.49 | 75.05 | 94.84 | 26.62 | 94.57 |
| **Energy** | 75.05 | 94.84 | 40.14 | 89.89 | 75.05 | 94.84 | 27.58 | 94.24 |
| **Mahalanobis** | 75.05 | 94.84 | 68.57 | 84.61 | 75.05 | 94.84 | 42.62 | 93.23 |
| **ViM** | 75.05 | 94.84 | 21.95 | 95.48 | 75.05 | 94.84 | 36.80 | 93.37 |
| **KNN** | 75.05 | 94.84 | 42.67 | 91.07 | 75.05 | 94.84 | 29.75 | 94.60 |
| **ASH** | 75.05 | 94.84 | 44.07 | 88.84 | 75.05 | 94.84 | 22.07 | 95.61 |
| *OOD generalization* | | | | | | | | |
| **ERM** | 75.05 | 94.84 | 40.14 | 89.89 | 75.05 | 94.84 | 27.58 | 94.24 |
| **Mixup** | 79.17 | 93.30 | 58.24 | 75.70 | 79.17 | 93.30 | 32.73 | 88.86 |
| **IRM** | 77.92 | 90.85 | 53.79 | 88.15 | 77.92 | 90.85 | 34.50 | 94.54 |
| **VREx** | 76.90 | 91.35 | 56.13 | 87.45 | 76.90 | 91.35 | 44.20 | 92.55 |
| **EQRM** | 75.71 | 92.93 | 51.00 | 88.61 | 75.71 | 92.93 | 31.23 | 94.94 |
| **SharpDRO** | 79.03 | 94.91 | 34.64 | 91.96 | 79.03 | 94.91 | 13.27 | 97.44 |
| *Learning w. $\mathbb{P}_{\text{wild}}$* | | | | | | | | |
| **OE** | 35.98 | 94.75 | 27.02 | 94.57 | 46.89 | 94.07 | 0.70 | 99.78 |
| **Energy (w/ outlier)** | 19.86 | 90.55 | 23.89 | 93.60 | 32.91 | 93.01 | 0.27 | 99.94 |
| **Woods** | 54.58 | 94.88 | 30.48 | 93.28 | 78.75 | 95.01 | 0.60 | 99.87 |
| **Scone** | 85.21 | 94.59 | 37.56 | 90.90 | 80.31 | 94.97 | 0.87 | 99.79 |
| **AHA (Ours)** | **88.93**$_{\pm0.06}$ | 94.30$_{\pm0.04}$ | **11.88**$_{\pm0.30}$ | **95.60**$_{\pm0.16}$ | **91.08**$_{\pm0.01}$ | 94.41$_{\pm0.00}$ | **0.07**$_{\pm0.00}$ | **99.98**$_{\pm0.00}$ |

**Places365** [113] is a large-scale image dataset comprising scene photographs. The dataset is divided into several subsets to facilitate the training and evaluation of scene classification. It is highly diverse and offers extensive coverage of various scene types.

**iNaturalist** [94] is a challenging real-world collection featuring species captured in diverse situations. It comprises 13 super-categories and 5,089 sub-categories. For our experiment, we use the subset provided by [48], which contains 110 plant classes with no overlap with the IMAGENET-1K categories [24].

**PACS** [64] is a commonly used OOD generalization dataset from DomainBed [40]. It includes four domains with different image styles: photo, art painting, cartoon, and sketch, and it covers seven categories. It is created by intersecting the classes found in Caltech256 (Photo), Sketchy (Photo, Sketch) [83], TU-Berlin (Sketch) [30], and Google Images (Art painting, Cartoon, Photo). This dataset consists of 9,991 examples with a resolution of $224 \times 224$ pixels.

**Data split details for OOD datasets and composing wild mixture data.** Following previous work [55, 6], we use different data splitting strategies for standard and OOD datasets. For datasets with a standard train-test split, such as SVHN, we use the original test split for evaluation. For other OOD datasets, we allocate 70% of the data to create the wild mixture training data and the mixture validation dataset, while the remaining 30% is reserved for test-time evaluation. Within the training/validation split, 70% of the data is used for training, and the remaining 30% is used for validation.

# E  Results of Additional OOD Datasets

Table 8 presents the main results on additional OOD datasets, including Places365 [113] and LSUN-Resize [103]. Our proposed approach achieves strong performance in OOD generalization and OOD detection on these datasets. We highlight some observations: (1) We compare our method with post-hoc OOD detection methods such as MSP [46], ODIN [67], Energy [68], Mahalanobis [62], ViM [98], KNN [91], and the most recent method ASH [25]. These approaches are all based on a model trained with cross-entropy loss, which demonstrates suboptimal OOD generalization performance. (2) We compare our method with OOD generalization approaches, including IRM [2], GroupDRO [81], Mixup [106], VREx [61], EQRM [29], and the most recent method SharpDRO [51]. Our approach achieves improved performance compared to these OOD generalization baselines. (3)

Additionally, we compare our method with learning from $\mathbb{P}_{\text{wild}}$ OOD baselines, such as OE [47], Energy [68], WOODS [55], and SCONE [6]. Our approach achieves strong performance on both OOD generalization and detection accuracy, demonstrating the effectiveness of our human-assisted OOD learning framework for both OOD generalization and OOD detection.

Table 9: Results on ImageNet-100. We use ImageNet-100 as ID, and iNaturalist for $\mathbb{P}_{\text{ood}}^{\text{semantic}}$.

| Method | OOD Acc.↑ | ID Acc.↑ | FPR95↓ | AUROC↑ |
|---|---|---|---|---|
| **WOODS** [55] | 44.46 | 86.49 | 10.50 | 98.22 |
| **SCONE** [6] | 65.34 | 87.64 | 27.13 | 95.66 |
| **AHA (Ours)** | **72.74** | 86.02 | **2.55** | **99.35** |

# F    Results on ImageNet-100

We provide additional large-scale results on the ImageNet benchmark. We use ImageNet-100 as the ID data ($\mathbb{P}_{\text{in}}$), with labels provided in Appendix D. For the semantic-shifted OOD data, we use the high-resolution natural images from iNaturalist [94], with the same subset as employed in the MOS approach [48]. We fine-tune a ResNet-34 model [43] (pre-trained on ImageNet) for 100 epochs, using an initial learning rate of 0.01 and a batch size of 64. Table 9 suggests that AHA can improve OOD detection performance compared to WOODS and SCONE, achieving better FPR95 and AUROC.

# G    Effect on different mixing ratios of wild data.

Table 10: Ablation on different mixing ratios of wild data. The labeling budget is $k = 1000$. The OOD score used is the energy score. We train on CIFAR-10 as ID, using wild data with $\pi_c$ (CIFAR-10-C) and $\pi_s$ (Texture).

| Ratios | Method | OOD Acc.↑ | ID Acc.↑ | FPR↓ | AUROC↑ |
|---|---|---|---|---|---|
| $\pi_c = 0.4$, | **Top-k** | 88.32 | 94.51 | 5.41 | 97.62 |
| $\pi_s = 0.3$ | **AHA (Ours)** | **89.46** | 94.50 | **3.19** | **98.83** |
| $\pi_c = 0.5$, | **Top-k** | 88.27 | 94.60 | 5.65 | 97.75 |
| $\pi_s = 0.2$ | **AHA (Ours)** | **88.72** | 94.38 | **4.75** | **98.48** |
| $\pi_c = 0.5$, | **Top-k** | 88.90 | 94.51 | 7.45 | 96.79 |
| $\pi_s = 0.1$ | **AHA (Ours)** | **89.58** | 94.73 | **6.37** | **97.26** |
| $\pi_c = 0.6$, | **Top-k** | 89.12 | 94.50 | 8.11 | 96.77 |
| $\pi_s = 0.1$ | **AHA (Ours)** | **89.31** | 94.59 | **7.33** | **96.88** |

In Table 10, we provide an ablation study on different fractions of covariate OOD $\pi_c$ and fractions of semantic OOD data $\pi_s$ within the wild distribution $\mathbb{P}_{\text{wild}}$. We focus primarily on evaluations where $\pi_c \neq 0$, $\pi_s \neq 0$, and $1 - \pi_c - \pi_s \neq 0$, as our problem uniquely introduces these three types of distributions in the wild. We observe that OOD generalization performance for top-$k$ sampling generally increases with a higher fraction of covariate OOD and a lower fraction of semantic OOD, since more covariate OOD are selected and annotated in top-$k$ sampling. Additionally, AHA consistently achieve better performance compared to top-$k$ for both OOD generalization and detection. The improvement is more significant with a larger fraction of semantic OOD data.

# H    Additional Visualization Results for Real Wild Data

We provide additional visualization on the OOD score distribution for various datasets in Figure 3.

# I    Hyperparameter Analysis

Table 11 provides an ablation study on varying the hyperparameter $\alpha$, which balances the weight between the two loss terms. We observe that the performance is strong and remains insensitive across a wide range of $\alpha$ values.

# J    Results of Different Covariate Data Types

We provide additional ablation studies of different covariate shifts (see Table 12). We evaluate AHA under 19 different common corruptions, including Gaussian noise, impulse noise, brightness, zoom

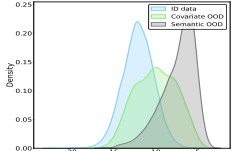 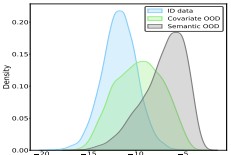 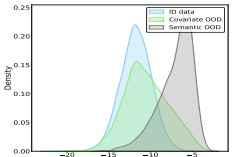 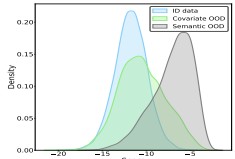

(a) Scores of $\mathbb{P}_{in}$ (CIFAR-10), $\mathbb{P}_{out}^{covariate}$ (Gaussian), $\mathbb{P}_{out}^{semantic}$ (SVHN).

(b) Scores of $\mathbb{P}_{in}$ (CIFAR-10), $\mathbb{P}_{out}^{covariate}$ (Gaussian), $\mathbb{P}_{out}^{semantic}$ (Places).

(c) Scores of $\mathbb{P}_{in}$ (CIFAR-10), $\mathbb{P}_{out}^{covariate}$ (Impulse noise), $\mathbb{P}_{out}^{semantic}$ (SVHN).

(d) Scores of $\mathbb{P}_{in}$ (CIFAR-10), $\mathbb{P}_{out}^{covariate}$ (Impulse noise), $\mathbb{P}_{out}^{semantic}$ (Places).

Figure 3: (a)-(e) Score distributions for the real wild data. Different colors represent the different types of test data: CIFAR-10 as $\mathbb{P}_{in}$ (blue), CIFAR-10-C as $\mathbb{P}_{out}^{covariate}$ (green), and Textures as $\mathbb{P}_{out}^{semantic}$ (gray).

Table 11: Ablation study on the effect of loss weight $\alpha$. The sampling strategy is top-$k$ sampling, with a budget of 1000. We train on CIFAR-10 as ID, using wild data with $\pi_c = 0.5$ (CIFAR-10-C) and $\pi_s = 0.1$ (Texture).

| Balancing weights | OOD Acc.↑ | ID Acc.↑ | FPR↓ | AUROC↑ |
|---|---|---|---|---|
| $\alpha$=1.0 | 90.01 | 94.53 | 3.25 | 98.98 |
| $\alpha$=3.0 | 89.80 | 94.51 | 3.19 | 98.83 |
| $\alpha$=5.0 | 89.73 | 94.53 | 3.19 | 89.73 |
| $\alpha$=7.0 | 89.66 | 94.55 | 3.25 | 98.99 |
| $\alpha$=9.0 | 89.59 | 94.51 | 3.19 | 99.05 |

blur, and others. These covariate shifts are generated based on previous literature [45]. Our approach is consistant performance under different covariate shifts and achieves enhanced OOD generalization and OOD detection performance.

## K  Software and Hardware

Our framework was implemented using PyTorch 2.0.1. Experiments are performed using Tesla V100.

## L  Broader Impact and Limitations

By improving OOD generalization and detection for machine learning models, this work can help increase the robustness and reliability of deployed AI systems across many real-world applications. Failing to properly handle distribution shifts is a key vulnerability of current AI that can lead to errors, discriminatory behavior, and safety risks. Our human-assisted framework leverages a strategic data labeling approach to cost-effectively boost OOD performance. This could benefit high-stakes domains like medical diagnosis, autonomous vehicles, financial services, and content moderation systems where distribution shifts are common and errors can have severe consequences.

At the same time, there are potential negative impacts to consider. While human labeling can improve model performance, it also introduces privacy risks if the labeling process exposes sensitive data. There are also potential risks of labeling bias or low quality labels degrading rather than enhancing model behavior. From an ethical AI perspective, increasing the capability and deployment of highly capable AI systems carries inherent societal risks that must be weighed against the benefits.

To mitigate these risks, we advocate for implementing robust data governance policies, secure data pipelines, anti-bias monitoring, and careful vetting of crowdsourced labels. We also encourage developing complementary approaches to make models more inherently robust to distribution shifts, rather than relying solely on human labeling which can be costly and difficult to scale. Overall, we believe the potential positive impacts of safer, more reliable AI outweigh the risks if appropriate safeguards are put in place. But we must remain vigilant about responsibly developing and deploying AI capabilities that are beneficial to society.

**Limitations.** While our human-assisted framework shows promising results for improving OOD generalization and detection, it still requires some human annotation effort. Further reducing the required labeling cost can be a key focus for future work.

Table 12: Ablations on the different covariate shifts. We train on CIFAR-10 as ID, using CIFAR-10-C as $\mathbb{P}_{\text{ood}}^{\text{covariate}}$ and SVHN as $\mathbb{P}_{\text{ood}}^{\text{semantic}}$ (with $\pi_c = 0.5$ and $\pi_s = 0.1$).

| Covariate shift type | Method | OOD Acc.↑ | ID Acc.↑ | FPR↓ | AUROC↑ |
|---|---|---|---|---|---|
| Gaussian noise | **WOODS** | 52.76 | 94.86 | 2.11 | 99.52 |
|  | **SCONE** | 84.69 | 94.65 | 10.86 | 97.84 |
|  | **AHA (Ours)** | 89.14 | 94.64 | 0.09 | 99.97 |
| Defocus blur | **WOODS** | 94.76 | 94.99 | 0.88 | 99.83 |
|  | **SCONE** | 94.86 | 94.92 | 11.19 | 97.81 |
|  | **AHA (Ours)** | 94.74 | 94.70 | 0.05 | 99.98 |
| Frosted glass blur | **WOODS** | 38.22 | 94.90 | 1.63 | 99.71 |
|  | **SCONE** | 69.32 | 94.49 | 12.80 | 97.51 |
|  | **AHA (Ours)** | 80.49 | 94.48 | 0.20 | 99.93 |
| Impulse noise | **WOODS** | 70.24 | 94.87 | 2.47 | 99.47 |
|  | **SCONE** | 87.97 | 94.82 | 9.70 | 97.98 |
|  | **AHA (Ours)** | 92.17 | 94.71 | 0.07 | 99.97 |
| Shot noise | **WOODS** | 70.09 | 94.93 | 3.73 | 99.26 |
|  | **SCONE** | 88.62 | 94.68 | 10.74 | 97.85 |
|  | **AHA (Ours)** | 91.87 | 94.56 | 0.07 | 99.97 |
| Snow | **WOODS** | 88.10 | 95.00 | 2.42 | 99.54 |
|  | **SCONE** | 90.85 | 94.83 | 13.22 | 97.32 |
|  | **AHA (Ours)** | 92.85 | 94.72 | 0.07 | 99.97 |
| Zoom blur | **WOODS** | 69.15 | 94.86 | 0.38 | 99.91 |
|  | **SCONE** | 90.87 | 94.89 | 7.72 | 98.54 |
|  | **AHA (Ours)** | 92.04 | 94.53 | 0.07 | 99.98 |
| Brightness | **WOODS** | 94.86 | 94.98 | 1.24 | 99.77 |
|  | **SCONE** | 94.93 | 94.97 | 1.41 | 99.74 |
|  | **AHA (Ours)** | 94.77 | 94.77 | 0.05 | 99.98 |
| Elastic transform | **WOODS** | 87.89 | 95.04 | 0.37 | 99.92 |
|  | **SCONE** | 91.01 | 94.88 | 8.77 | 98.32 |
|  | **AHA (Ours)** | 90.99 | 94.74 | 0.07 | 99.97 |
| Contrast | **WOODS** | 94.37 | 94.94 | 1.06 | 99.80 |
|  | **SCONE** | 94.40 | 94.98 | 1.30 | 99.77 |
|  | **AHA (Ours)** | 94.34 | 94.66 | 0.06 | 99.98 |
| Fog | **WOODS** | 94.69 | 95.01 | 1.06 | 99.80 |
|  | **SCONE** | 94.71 | 95.00 | 1.35 | 99.76 |
|  | **AHA (Ours)** | 94.67 | 94.72 | 0.07 | 99.98 |
| Frost | **WOODS** | 87.25 | 94.97 | 2.35 | 99.55 |
|  | **SCONE** | 91.94 | 94.85 | 10.08 | 98.03 |
|  | **AHA (Ours)** | 92.23 | 94.73 | 0.05 | 99.98 |
| Gaussian blur | **WOODS** | 94.78 | 94.98 | 0.87 | 99.83 |
|  | **SCONE** | 94.76 | 94.86 | 3.14 | 99.39 |
|  | **AHA (Ours)** | 94.58 | 94.72 | 0.05 | 99.98 |
| Jpeg | **WOODS** | 84.35 | 94.96 | 1.73 | 99.68 |
|  | **SCONE** | 87.87 | 94.90 | 8.14 | 98.49 |
|  | **AHA (Ours)** | 89.24 | 94.53 | 0.05 | 99.98 |
| Motion blur | **WOODS** | 82.54 | 94.79 | 0.47 | 99.88 |
|  | **SCONE** | 91.95 | 94.90 | 9.15 | 98.18 |
|  | **AHA (Ours)** | 92.58 | 94.59 | 0.05 | 99.98 |
| Pixelate | **WOODS** | 91.56 | 94.91 | 1.82 | 99.66 |
|  | **SCONE** | 92.08 | 94.96 | 1.97 | 99.64 |
|  | **AHA (Ours)** | 93.34 | 94.61 | 0.05 | 99.98 |
| Saturate | **WOODS** | 92.45 | 95.03 | 1.26 | 99.77 |
|  | **SCONE** | 93.38 | 94.92 | 10.27 | 97.88 |
|  | **AHA (Ours)** | 93.40 | 94.79 | 0.06 | 99.98 |
| Spatter | **WOODS** | 92.38 | 94.98 | 1.94 | 99.64 |
|  | **SCONE** | 92.78 | 94.98 | 1.94 | 99.64 |
|  | **AHA (Ours)** | 93.68 | 94.73 | 0.07 | 99.97 |
| Speckle noise | **WOODS** | 72.31 | 94.94 | 3.51 | 99.30 |
|  | **SCONE** | 88.51 | 94.83 | 11.05 | 97.82 |
|  | **AHA (Ours)** | 92.00 | 94.73 | 0.08 | 99.97 |
| **Average** | **WOODS** | 81.72 | 94.94 | 1.65 | 99.68 |
|  | **SCONE** | 90.29 | 94.86 | 7.62 | 98.50 |
|  | **AHA (Ours)** | **92.06** | 94.67 | **0.07** | **99.97** |

