# OpenReview forum: "AHA: Human-Assisted Out-of-Distribution Generalization and Detection"
_NeurIPS.cc/2024/Conference — NeurIPS 2024 poster_

### Official Review · Reviewer_CyPp · 2024-06-13

**Soundness:** 3
**Presentation:** 3
**Contribution:** 3
**Rating:** 7
**Confidence:** 4

**Summary:**

The paper presents a novel approach to the problem of OOD generalization and detection. The authors introduce the AHA (Adaptive Human-Assisted OOD learning) framework, which aims to enhance both out-of-distribution (OOD) generalization and detection by strategically leveraging human-assisted labeling within a maximum disambiguation region. The paper reports significant improvements over state-of-the-art methods with only a few hundred human annotations, demonstrating the efficacy of the proposed framework.

**Strengths:**

- The AHA framework is a creative solution that addresses the challenges of OOD generalization and detection, which are critical for real-world applications of machine learning models.
- The authors provide extensive experimental results that validate the effectiveness of their approach, showing robust performance across various datasets.
- The incorporation of human feedback in a strategic manner is a strength, as it capitalizes on the limited labeling budget to maximize model performance.
- The paper's contributions are articulated, with the novel labeling strategy and the integration of human assistance being the highlights.
- The transformation of the problem into a noisy binary search is an intelligent methodological choice that allows for the efficient identification of the maximum ambiguity threshold.

**Weaknesses:**

- While the paper demonstrates strong results, it is not clear how the AHA framework scales with larger and more complex datasets.
- The reliance on human annotations could be a limitation in scenarios where such resources are not readily available or are cost-prohibitive.
- The paper could benefit from a discussion on how the findings generalize beyond the tested datasets and scenarios.
- The computational complexity of the AHA algorithm and its runtime performance on large datasets are not discussed.
- The paper could address potential biases introduced by human labeling, especially in the context of OOD detection.

**Questions:**

- How does the AHA framework perform as the size and complexity of the dataset increase?
- What are the specific steps taken to mitigate potential biases in human labeling?
- Can the authors provide more details on the computational efficiency of the AHA algorithm, especially for large-scale applications?
- How does the framework handle a class imbalance in the context of OOD detection?
- Are there any specific domains or applications where the AHA framework is expected to be more or less effective, and why?
- Could the proposed method benefit the OOD detection with unreliable sources [R1] and inspire unsupervised OOD detection [R2]?

----
[R1] Out-of-distribution detection learning with unreliable out-of-distribution sources. NeurIPS 2023.
[R2] Out-of-distribution detection with an adaptive likelihood ratio on informative hierarchical vae. NeurIPS 2022.

**Limitations:**

NA.

---

> ### Author Rebuttal · Authors · 2024-08-06
>
> We thank you for the detailed comments and questions, which we address in detail below.
>
> > *W1. While the paper demonstrates strong results, it is not clear how the AHA framework scales with larger and more complex datasets.*
>
> We tested the AHA framework on the larger and more complex ImageNet benchmark. We use ImageNet-100 as the ID data, ImageNet-100-C with Gaussian noise as the covariate OOD data, and high-resolution natural images from iNaturalist for the semantic OOD data. Our AHA method still achieves much better performance in OOD detection in terms of FPR95 and AUROC compared to WOODS and SCONE.
>
> Results for both OOD generalization and OOD detection are summarized below:
>
> | Method | OOD Accuracy | ID Accuracy | FPR95 | AUROC |
> | -------- | -------- | -------- |-------- |-------- |
> | WOODS | $44.46$ | $86.49$ | $10.50$ | $98.22$ |
> | SCONE | $65.34$ | $87.64$ | $27.13$ | $95.66$ |
> | AHA (Ours) | **72.74** | $86.02$ | **2.55** | **99.35** |
>
> > *W2. The reliance on human annotations could be a limitation in scenarios where such resources are not readily available or are cost-prohibitive.*
>
> This is a valid point. There may indeed be special scenarios where resources are not available or are cost-prohibitive. We will include this in the limitations discussion section. However, our proposed AHA framework is label-efficient, requiring only a small portion of labels (1\% $\sim$ 2\%) to be effective, which significantly reduces the cost associated with human annotation requirements.
>
> > *W3. The paper could benefit from a discussion on how the findings generalize beyond the tested datasets and scenarios.*
>
> We thank you for the suggestion. The findings may not generalize to special scenarios where there is no clear boundary between covariate-shift and semantic-shift data, and these two types of OOD samples can be very hard to separate. In such cases, the proposed AHA method might downgrade and become comparable to random sampling. However, in most real-world scenarios, there are usually discernible differences between covariate and semantic shifts.
>
> > *W4. The computational complexity of the AHA algorithm and its runtime performance on large datasets are not discussed.*
>
> The computational complexity of the AHA algorithm is $O(Nk)$, where $N$ denotes the number of examples and $k$ denotes the budget size. The average runtime for the core part of the noisy binary search process in the AHA algorithm is 0.3232 milliseconds per point. For a large dataset, for example when the dataset size is 60k, the total runtime performance for the noisy binary search step is around 19 seconds. This step is inexpensive compared to neural network training.
> For the last step, training of the multi-class classifier and OOD detector with selected and annotated examples based on the loss objective in Section 4.4 usually takes several hours to converge.
>
> > *W5. The paper could address potential biases introduced by human labeling, especially in the context of OOD detection.*
>
> Here are the specific steps taken to mitigate potential biases in human labeling:
> We propose using diverse labelers and providing thorough training for them on recognizing and avoiding biases. Additionally, we will implement validation procedures to ensure label quality, such as having a subset of data labeled by multiple people and utilizing expert reviewers. These steps could help to reduce the potential biases introduced by human labeling.
>
> Moreover, our framework is label-efficient and only requires 1\% $\sim$ 2\% annotations for the wild data. This makes it practical for manual double verification to address potential biases and ensure label quality, even in cost-prohibitive scenarios.
>
> > *Q4. How does the framework handle a class imbalance in the context of OOD detection?*
>
> Our framework uses metrics such as AUROC, which are less sensitive to class imbalance and provide a more accurate assessment of performance even in imbalanced scenarios.
>
> > *Q5. Are there any specific domains or applications where the AHA framework is expected to be more or less effective, and why?*
>
> Referring to our response to W3, the AHA framework is expected to be less effective in rare scenarios where there is no clear boundary between covariate OOD and semantic OOD data, as these two types of OOD samples can be very hard to separate. We will add this discussion to the main paper.
>
> > *Q6. Could the proposed method benefit the OOD detection with unreliable sources [R1] and inspire unsupervised OOD detection [R2]?*
>
> This could be part of our future work to extend the AHA framework by considering unreliable sources and unsupervised OOD detection. We will cite and incorporate these two literature in the future work discussion section.

---

> > ### Comment · Reviewer_CyPp · 2024-08-08
> >
> > Thanks for your feedback. It addresses my concerns, especially experiments on large-scale datasets. I have decided to increase the rating on soundness and my confidence. The need for human assistance is both the motivation of this work and its inevitable weakness, depending on the task requirements and the specific scenario. Overall, I am positive about this work (weak acceptance). However, I also would like to hear other reviewers' opinions and discuss this, and make further judgments.

---

> > > ### Author Response · Authors · 2024-08-08
> > >
> > > We thank you for reading our response and your positive feedback.

---

> > > > ### Author Response · Authors · 2024-08-13
> > > > **Response to Reviewer CyPp (Followup)**
> > > >
> > > > Thank you again for your feedback and comments. We address your comments below.
> > > >
> > > > > *The need for human assistance is both the motivation of this work and its inevitable weakness.*
> > > >
> > > > We would like to clarify that there are many scenarios where human annotation is particularly valuable, for example, in medical diagnostics. In these contexts, human assistance provides crucial expertise and contextual information, which should be considered a significant advantage rather than a weakness.
> > > >
> > > > We would greatly appreciate an increased overall rating if we have successfully answered your questions. Otherwise, we are happy to provide additional discussions to address any further concerns.

---

### Official Review · Reviewer_Rqpf · 2024-06-24

**Soundness:** 3
**Presentation:** 3
**Contribution:** 2
**Rating:** 5
**Confidence:** 4

**Summary:**

This paper proposes to address both out-of-distribution detection and generalization within one joint framework under human-assistance. The proposed method first utilizes a noisy binary search algorithm to identify the most informative samples to be labeled. Then, it continues to annotate these samples with human feedback. The authors conduct experiments on CIFAR and PACS to evaluate the proposed method.

**Strengths:**

- The proposed method can handle OOD detection and generalization at the same time, which is impactful to both these two individual research areas.
- Covariat-shifts and semantic shifts are both inevitable in real-world applications. Thus the proposed method is straightforward and well-motivated.
- Most parts of this paper are well presented with good visualization.

**Weaknesses:**

- AHA may only work under a rather strict assumption. Compared to outlier exposure, AHA needs to access the real test data distribution $S_{wild}$ to selectively label some samples. While outlier exposure does not need such an assumption. Many previous OOD detection methods can only access training data distribution and an auxiliary OOD data distribution (noted that such auxiliary OOD has no overlapped samples with test-time OOD data in common settings). Thus I think AHA may only work under a more strict assumption (i.e., the test data distribution $S_{wild}$ is accessible) than previous outlier exposure.
- I generally believe enhancing OOD detection and generalization with human feedback is laborious. I am aware the proposed method can get good performance with hundreds or thousands of labeled samples in many cases. However, such human feedback still seems laborious to me. For example, in CIFAR experiments, the images are only 32*32. Thus I think it would take a lot of time to label such samples. Not to mention the samples can be noised or corrupted (CIFAR-10-C).
- The experiments are not adequate. It is widely acknowledged that OOD detection and generalization are more difficult on large-scale high-resolution datasets. For example, ImageNet benchmarks. The authors do not conduct evaluations on such datasets.
- In some cases, there may be no clear boundary between covariate-shift data and semantic shift data. As mentioned in recent work[1], these two types of OOD samples can be very hard to separate (even for humans). Could the authors comment on this phenomenon?

Given the above points, I tend to reject this paper because the overall quality do not meet the expectations of NIPS. However, I may adjust my score if there is a strong argument.

[1] Yang, William, Byron Zhang, and Olga Russakovsky. "ImageNet-OOD: Deciphering Modern Out-of-Distribution Detection Algorithms."

**Questions:**

Please refer to weaknesses and address my concerns.

---

> ### Author Rebuttal · Authors · 2024-08-06
>
> We thank you for your thorough comments and questions, which we address in detail below.
>
> > *W1. AHA may only work under a rather strict assumption.*
>
> We would like to clarify that the unlabeled wild data distribution $S_{\text{wild}}$ is totally different from the test distribution. Our driving motivation is to exploit unlabeled wild data with various distribution shifts naturally arising in real-world scenarios. Thus, we consider the following generalized characterization of the wild data to model the realistic environment:
>
> $P_\text{wild}:= (1-\pi_s-\pi_c) P_\text{in} + \pi_c P_\text{out}^\text{covariate} +\pi_s P_\text{out}^\text{semantic}$
>
> Moreover, outlier exposure has a more strict assumption of careful data cleaning to ensure the auxiliary outlier data does not overlap with the ID data. Compared to outlier exposure, we relax this assumption by leveraging unlabeled "in-the-wild" data, which is a mixture of ID, covariate OOD, and semantic OOD data commonly observed in real-world applications.
>
> > *W2. I generally believe enhancing OOD detection and generalization with human feedback is laborious.*
>
> Our method significantly reduces the labeling effort compared to traditional approaches. With no more than 2\% annotations of the wild data, our method outperforms existing state-of-the-art methods by reducing OOD detection error by 15.79\% and increasing OOD generalization accuracy by 5.05\%.
>
> Regarding the CIFAR experiments, while the images are small (32x32), our method only requires binary "In" vs. "Out" labels for the selected semantic OOD data, and category labels for the covariate OOD data. This simple classification task is much faster than providing detailed labels or descriptions, significantly reducing the time and effort needed from human annotators.
>
> We also note that labeling a few hundred OOD samples is generally much cheaper than labeling in-distribution examples. For reference, labeling 200 images with five labelers each costs less than $10 on Amazon Mechanical Turk. We believe the trade-off between this minimal labeling effort and the substantial performance improvement, especially in challenging OOD scenarios, makes our approach particularly efficient and practical for real-world applications.
>
> > *W3. The experiments are not adequate. It is widely acknowledged that OOD detection and generalization are more difficult on large-scale high-resolution datasets.*
>
> The experiments on large-scale, high-resolution datasets are included in **Appendix G**. Following the ImageNet benchmark for joint OOD generalization and OOD detection as used in SCONE literature [1], we use ImageNet-100 as the in-distribution data, with labels details provided in **Appendix E**. For the covariate OOD data, we use ImageNet-100-C with Gaussian noise in the experiment. For the semantic OOD data, we use the high-resolution natural images from iNaturalist.
>
> Results for both OOD generalization and OOD detection evaluation are summarized below:
>
> | Method | OOD Accuracy | ID Accuracy |FPR95 |AUROC |
> | -------- | -------- | -------- |-------- |-------- |
> | WOODS | $44.46$ | $86.49$ | $10.50$ | $98.22$ |
> | SCONE | $65.34$ | $87.64$ | $27.13$ | $95.66$ |
> | AHA (Ours) | **72.74** | $86.02$ | **2.55** | **99.35** |
>
> These experiments demonstrate that our method maintains its effectiveness on more complex datasets.
>
> [1] Feed Two Birds With One Scone: Exploiting Wild Data for Both Out-of-Distribution Generalization and Detection. ICML 2023.
>
> > *W4. In some cases, there may be no clear boundary between covariate-shift data and semantic shift data.*
>
> We acknowledge the valid point raised about the potential difficulty in distinguishing between covariate-shift and semantic-shift data in some cases. However, in such scenarios, our proposed AHA method would naturally default to unbiased sampling, performing no worse than a random sampling strategy.
>
> Moreover, it's important to note that such cases are relatively rare in practice. Most real-world scenarios do exhibit discernible differences between covariate and semantic shifts, as verified empirically in **Section 5.4** of our main paper.

---

> > ### Comment · Reviewer_Rqpf · 2024-08-08
> >
> > I thank the authors for their detailed explanation in the rebuttal. My concern about performance on high-resolution large-scale benchmarks has been addressed (w3). The argument for w2 is also OK for me. I recommend adding these to the paper to enhance the transparency about the cost of additional labeling. However, I still have further concerns about the basic settings of this paper. AHA assumes that there is an overlap between test-time covariant-shifted data distribution and training-time unlabeled data. For instance, it assumes that during training, the model can access samples corrupted by Gaussian noise. It is wired to me and may be a rather strict condition compared to that used in standard OOD generalization methods. What if we keep the test-time covariant-shifted distribution unknown? For example, only expose the model to one certain type of corruption, but test it on another type of corruption.

---

> ### Author Response · Authors · 2024-08-09
> **Response to Reviewer Rqpf (Followup)**
>
> Thank you for taking the time to read our response. We address your comments below.
>
> > *1. I still have further concerns about the basic settings of this paper.*
>
> We would like to clarify that the wild data setting, which includes a mixture of ID, covariate OOD, and semantic OOD data, is commonly observed in practice. The overall wild mixture data distribution $P_\text{wild}$ differs from the test environment data distribution. Additionally, the mixing ratios of $π_s$ and $π_c$ are unknown in our formulation, making this setting well-suited to real-world scenarios with varied distributions of wild data. We have also empirically tested different mixing ratios, as detailed in **Appendix H**, where we demonstrate the robust and strong performance of our AHA framework.
>
> Moreover, as suggested, we have summarized the OOD accuracy results on covariate test data when the model is exposed to one type of corruption but tested on another. Specifically, we exposed the model to wild data containing Gaussian noise corruption and then tested it on nine other types of corruption: impulse noise, spatter, shot noise, saturate, speckle noise, frosted glass blur, motion blur, frost, and zoom blur. The results indicate AHA displays strong performance, even when the test-time covariate-shifted distribution remains unknown.
>
>
> | Algorithm | Impulse noise |      Spatter     | Shot noise |  Saturate  | Speckle noise | Frosted glass blur |   Motion blur   |    Frost   |   Zoom blur  |   Average    |
> | --------- | :-------------: | :----------------: | :----------: | :----------: | :-------------: | :---------: | :------------------: | :---------------: | :----------: | :----------: |
> | ERM       |    $85.15$    |     $92.88$      |   $84.62$  |  $92.49$   |     $84.46$   |            $52.29$        |    $89.10$      |  $89.67$   |   $85.55$    |   $84.02$    |
> | AHA (ours)|  **87.25**    |     **93.03**    |  **90.90** | **92.72**  |    **90.77**  |    **63.85**       |   **89.35**     | **91.52**  |  **86.58**   |  **87.33**   |
>
>
>
>
> > *2. Regarding the argument for W2.*
>
> Thank you for your constructive comments. We will incorporate discussions about the cost of additional labeling in the paper as suggested.

---

> > ### Author Response · Authors · 2024-08-11
> >
> > Thank you again for your constructive comments. If we've successfully resolved your previous concerns, we would appreciate an improved score. If you have any further comments, please feel free to let us know. We're happy to discuss and address any concerns.

---

> ### Comment · Reviewer_Rqpf · 2024-08-12
>
> Thanks for your response and hard work during the rebuttal. I am pleased to acknowledge that this paper has no significant flaws and the additional experiments are well-appreciated. However, I agree with other reviewers that the need for human assistance is both an advantage and an inevitable weakness of AHA. The uncommon settings should also be carefully explained and compared with classic domain adaption or OOD generalization. It seems to be a rather strict assumption than those used in domain adaption or OOD generalization (accessing samples drawn from test-time covariant-shift distribution and labeling some of them). Removing such an assumption can greatly strengthen the proposed method (e.g., involving various types of corruption). Theoretical support of why unlabeled data can help both OOD detection and generalization in AHA can be also taken into consideration in future work. With a better understanding of this paper, I have adjusted my score accordingly.

---

> > ### Author Response · Authors · 2024-08-12
> >
> > Thank you for your insightful feedback and constructive comments, which have been invaluable in enhancing our manuscript. We will incorporate the additional results (including various types of corruptions) and discussions in the paper as suggested. Regarding the need for human assistance, there are many applications where human annotation is particularly useful, such as medical diagnostics. In these contexts, human assistance provides crucial expertise and contextual information, and should be considered a significant advantage rather than a limitation.

---

### Official Review · Reviewer_Eoxk · 2024-07-13

**Soundness:** 3
**Presentation:** 3
**Contribution:** 3
**Rating:** 5
**Confidence:** 4

**Summary:**

This paper introduces a novel, integrated approach AHA (Adaptive Human-Assisted OOD learning) to simultaneously address both OOD generalization and detection through a human-assisted framework by labeling data in the wild. Extensive experiments validate the efficacy of AHA.

**Strengths:**

1. this paper is well written and easy to follow
2. good visualization and extensive experiments
3. Maximum Disambiguation Region and reduction to noisy binary search are new to me

**Weaknesses:**

1. there exsits a strong assumption that the weighted densities of semantic and covariance ood should equalize
2.  what is difference between active learning and the proposed Human-Assisted Learning
3. what is the time used for noisy binary search
4. whether the lamda searched on the one dataset can be transfer to another dataset

**Questions:**

see weakness

**Limitations:**

see weakness

---

> ### Author Rebuttal · Authors · 2024-08-06
>
> We thank you for your positive feedback and comments. We address each comment below in detail.
>
> > *W1. There exists a strong assumption that the weighted densities of semantic and covariance ood should equalize*
>
> Thank you for pointing out this potential misunderstanding. We would like to clarify that our formulation does not require the strong assumption that the weighted densities of semantic and covariate OOD should be equal. This holds for both our maximum disambiguation region formulation (Section 4.2) and the AHA algorithm (Section 4.3).
>
> Specifically, our maximum ambiguity threshold formulation in Eq. (1):
>
> $\lambda_* = argmax_{\mu \in R} \int_0^\mu ((1-\pi_c -\pi_s)p_{\text{in}}(\nu) + \pi_c p_{\text{covariate}}(\nu)) - \pi_s p_{\text{semantic}}(\nu) d\nu$
>
> indicates that if the density of covariate OOD is much smaller than semantic OOD, we would choose the smallest OOD scoring value. Conversely, if the density of covariate OOD is much larger than semantic OOD, we would choose the largest OOD scoring value, where we try to sample as many semantic OOD examples as possible.
>
>
> > *W2. What is difference between active learning and the proposed Human-Assisted Learning*
>
> Human involvement in labeling data points has been studied under various terminologies, including human-in-the-loop learning, bandits, interactive learning, and active learning. The title "Human-Assisted Learning" stems from our finding that human assistance can dramatically improve OOD generalization and detection performance.
>
> We provide discussions on active learning in **Appendix C**. To offer more context here, classic active learning involves an iterative training process, while our proposed algorithm only requires a single model fine-tuning. Additionally, existing deep active learning works do not study OOD robustness and the challenges posed by realistic scenarios involving wild data. Our proposed AHA method is specifically tailored for both OOD generalization and detection challenges.
>
>
> > *W3. What is the time used for noisy binary search*
>
> The average time for the noisy binary search process is 0.3232 milliseconds per point on Tesla V100 GPU.
>
> > *W4. Whether the lamda searched on the one dataset can be transfer to another dataset*
>
> The lambda searched on one dataset is not transferable to another dataset due to variations in data distributions. While lambda itself can't be transferred, the process of searching for the optimal lambda for each dataset is efficient and not computationally expensive. For example, the processing time for a dataset with 60k samples is about 19 seconds when using the Tesla V100 GPU.

---

> > ### Comment · Reviewer_Eoxk · 2024-08-09
> > **Official Comment by Reviewer Eoxk**
> >
> > Thank the authors for answering my questions. I would like to keep my rating "Borderline accept".

---

> > > ### Author Response · Authors · 2024-08-12
> > >
> > > Thank you again for your positive evaluation and feedback. If we have successfully addressed your questions, we would greatly appreciate an improved score. Otherwise, we are more than willing to provide additional discussions to address any further concerns.

---

### Comment · Area_Chair_AV1R · 2024-08-08
**The discussion has begun**

Dear Reviewers,

Thank you for taking the time to review the paper. The discussion has begun, and active participation is highly appreciated and recommended.

Thanks for your continued efforts and contributions to NeurIPS 2024.

Best regards,

Your Area Chair

---

### Decision · Program_Chairs · 2024-09-25

**Decision:**

Accept (poster)

**Comment:**

This paper introduces a novel approach AHA (Adaptive Human-Assisted OOD learning) to simultaneously address both OOD generalization and detection through a human-assisted framework by labeling data in the wild. The proposed approach strategically labels examples within a novel maximum disambiguation region, where the number of semantic and covariate OOD data roughly equalizes. Reviewers agree that this paper is novel to this field and might be an interesting research direction in the field of OOD detection. Key technical questions are addressed well during the rebuttal, and no technical flaws exist.